# O-GlcNAcylation of SIRT1 enhances its deacetylase activity and promotes cytoprotection under stress

Cuifang Han[1,3,4], Yuchao Gu [1,2,3,4], Hui Shan[1,2,3,4], Wenyi Mi[1,3,4], Jiahui Sun[1,2,4], Minghui Shi[1,2,4], Xinling Zhang[1,3,4], Xinzhi Lu[1,3,4], Feng Han [1,3,4], Qianhong Gong[1,3,4] & Wengong Yu[1,2,3,4]

SIRT1 is the most evolutionarily conserved mammalian sirtuin, and it plays a vital role in the regulation of metabolism, stress responses, genome stability, and ageing. As a stress sensor, SIRT1 deacetylase activity is significantly increased during stresses, but the molecular mechanisms are not yet fully clear. Here, we show that SIRT1 is dynamically modified with O-GlcNAc at Ser 549 in its carboxy-terminal region, which directly increases its deacetylase activity both in vitro and in vivo. The O-GlcNAcylation of SIRT1 is elevated during genotoxic, oxidative, and metabolic stress stimuli in cellular and mouse models, thereby increasing SIRT1 deacetylase activity and protecting cells from stress-induced apoptosis. Our findings demonstrate a new mechanism for the activation of SIRT1 under stress conditions and suggest a novel potential therapeutic target for preventing age-related diseases and extending healthspan.

[1] School of Medicine and Pharmacy, Ocean University of China, 5 Yushan Road, Qingdao 266003, China. [2] Laboratory for Marine Drugs and Bioproducts of Qingdao National Laboratory for Marine Science and Technology, Qingdao 266200, China. [3] Key Laboratory of Marine Drugs, Ministry of Education, Qingdao 266003, China. [4] Key Laboratory of Glycoscience & Glycotechnology of Shandong Province, Qingdao 266003, China. Cuifang Han, Yuchao Gu and Hui Shan contributed equally to this work. Correspondence and requests for materials should be addressed to Y.G. (email: guych@ouc.edu.cn) or to W.Y. (email: yuwg66@ouc.edu.cn)

The silent information regulator-2 (SIR2), or sirtuin, protein family is highly conserved from bacteria to humans[1], and its functions regarding longevity in yeast, *C. elegans* and *Drosophila* aroused academic and industrial interests[2–4]. There are seven different sirtuins, SIRT1-7, that have been identified in mammals, of which SIRT1 is the most closely related to SIR2[5]. SIRT1 is a $NAD^+$-dependent deacetylase[6] that belongs to the class III histone deacetylases. SIRT1 deacetylates and regulates histones as well as a wide range of non-histone substrates, including p53[7], forkhead (Fox) transcription factors[8], Ku70[9], peroxisome proliferator-activated receptor γ (PPARγ)[10], PPARγ coactivator-1α[11], nuclear factor kappa B[12], eukaryotic translation initiation factor 2α[13], heat shock factor 1[14], and others. By modulating these proteins, SIRT1 is implicated in a variety of cellular processes, including metabolism[15], DNA repair[16], genomic stability[17], cell cycle[8], cell survival and apoptosis[7], cellular senescence[18], and oncogenesis[19].

Organisms are able to activate genes responsible for cellular repair and protection in response to stress stimuli[20]. As one of the critical stress sensors, SIRT1 activity and levels are modulated by

**Fig. 1** OGT interacts with SIRT1 in vivo and in vitro. **a** NCI-H1299 whole-cell extracts (Input) and immunoprecipitates of anti-SIRT1 antibody and control IgG were analyzed by IB, with the SIRT1-silenced NCI-H1299 cells used as a negative control. Data represent two independent experiments. **b** CoIPs by anti-OGT antibody from either control or OGT-silenced NCI-H1299 whole-cell extracts were analyzed by IB. Data represent two independent experiments. **c** Co-localization of endogenous OGT (Alexa Fluor 594) and SIRT1 (Alexa Fluor 488) was determined by immunofluorescence in NCI-H1299 cells. DAPI staining was used to identify the nucleus. Scale bars: 20 μm. **d** Direct interaction of SIRT1 with GST-OGT in an in vitro GST pull-down assay. Data represent two independent experiments. **e** Mapping of the SIRT1 domain responsible for the interaction with OGT. GST pull-down assays were used to determine the interaction of OGT with GST-SIRT1 deletion mutants. Data represent two independent experiments

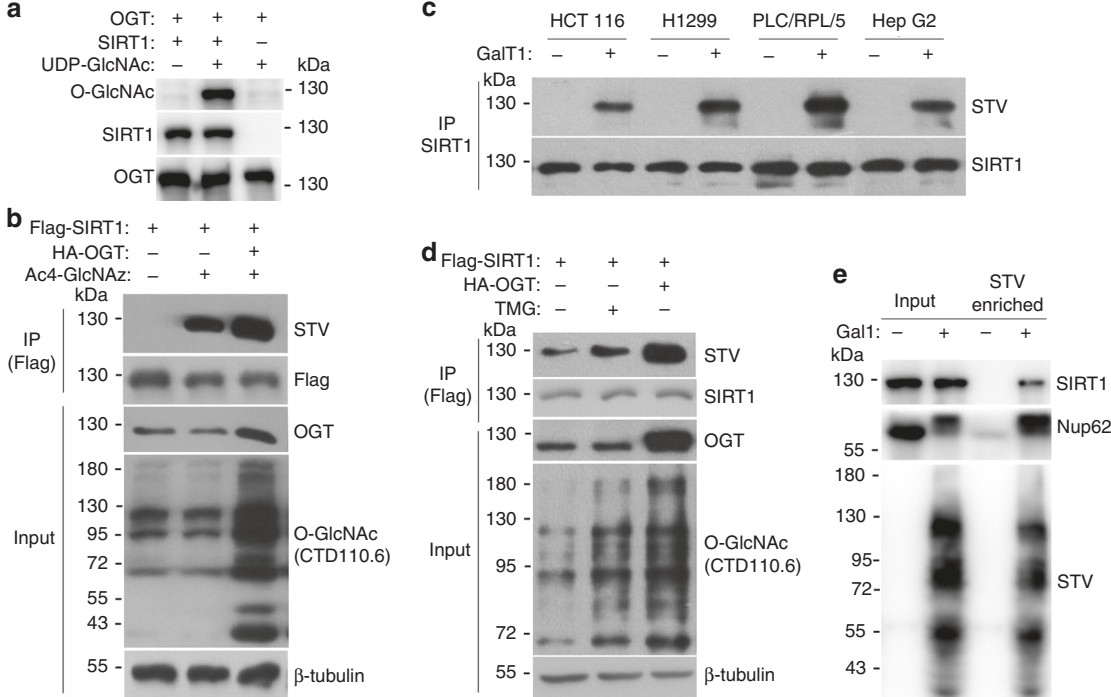

**Fig. 2** SIRT1 is O-GlcNAcylated in vitro and in vivo. **a** O-GlcNAcylation assays using recombinant human SIRT1 and OGT were performed in the presence or absence of UDP-GlcNAc. Reaction mixtures were immunoblotted with antibodies against O-GlcNAc (RL2), SIRT1, and OGT. Data represent two independent experiments. **b** Plasmids encoding Flag-SIRT1 and HA-OGT were transfected into NCI-H1299 cells, and the O-GlcNAcylation of SIRT1 was detected by either GlcNAz metabolic labeling with a biotin tag and probed with STV-HRP or the indicated antibodies. Data represent three independent experiments. **c** Detection of O-GlcNAcylated SIRT1 in the indicated cells by chemoenzymatic labeling. Cells were lysed, immunoprecipitated with antibody against SIRT1 and subsequently labeled with GalNAz and biotin, which were then probed with STV-HRP and antibody against SIRT1. Data represent two independent experiments. **d** Flag-tagged wtSIRT1 with or without HA-OGT was exogenously expressed in NCI-H1299 cells, and the cells were treated with or without 2 μM TMG for 4 h. Cells were lysed, immunoprecipitated with antibody against Flag-tag and subsequently labeled with GalNAz and biotin, which were then probed with STV-HRP and the indicated antibodies. Data represent three independent experiments. **e** NCI-H1299 cells, which were pretreated with 2 μM TMG for 4 h, were lysed and labeled with GalNAz and biotin. And then the biotinylated proteins were enriched and probed with indicated antibodies. Data represent two independent experiments

multiple cellular stresses, including genotoxic, oxidative, metabolic (e.g., calorie restriction), and proteotoxic stress, thus allowing for the coordination of the appropriate cellular response[21]. Therefore, it is important to clarify the mechanism by which the activity of SIRT1 is regulated under stress. Although it has been found that the NAD$^+$/NADH ratio[22], protein–protein interactions[23–25], and some post-translational modifications (PTMs)[26–28] are involved in the activation of SIRT1, a full mechanism of how enzyme activity is regulated under both normal and stress conditions remains unknown.

O-linked N-acetyl-β-D-glucosamine (O-GlcNAc) is a ubiquitous PTM on hydroxyl groups of serine and/or threonine residues of nuclear and cytoplasmic proteins. The O-GlcNAcylation of a protein and the removal of O-GlcNAc from a protein are catalyzed by O-GlcNAc transferase (OGT)[29] and a neutral-N-acetyl-β-glucosaminidase (O-GlcNAcase, OGA)[30], respectively. This dynamic O-GlcNAc cycling regulates diverse protein and cellular functions[31] as well as some diseases[32]. Similar to SIRT1, O-GlcNAc is also a stress sensor[33]. In response to various cellular stresses, global O-GlcNAcylation levels are increased[34] and, in turn, elevated O-GlcNAcylation appears to promote cell survival by participating in a multitude of biological processes, including the phosphoinositide 3-kinase/Akt pathway, heat shock protein expression, calcium homeostasis, levels of reactive oxygen species, ER stress, protein stability, mitochondrial dynamics, and inflammation[33, 35].

The overlapping roles of SIRT1 and O-GlcNAc in stress protection pathways prompted us to address the possibility that SIRT1 is modified with O-GlcNAc. Here, we show that SIRT1 is indeed O-GlcNAcylated, and this modification increases its deacetylase activity. Additionally, its O-GlcNAcylation level and deacetylase activity are enhanced under stress conditions. Exogenous expression of wild-type SIRT1 (wtSIRT1) promotes cell survival by increasing protein deacetylation, but the mutation of its O-GlcNAcylation site attenuates this activity.

## Results

**OGT directly binds SIRT1.** As OGT is the only known cytonuclear enzyme for intracellular protein O-GlcNAcylation, we first investigated whether OGT could bind to SIRT1 to evaluate the possible links between O-GlcNAcylation and SIRT1. NCI-H1299 cell extracts were co-immunoprecipitated with either an anti-SIRT1 antibody or a control IgG. As expected, immunoblotting (IB) assays revealed that OGT was clearly detected in the co-immunoprecipitates obtained with the anti-SIRT1 antibody but not with the control antibody (Fig. 1a). To confirm the specificity of the SIRT1 antibody, SIRT1 expressed in NCI-H1299 cells was silenced with SIRT1-specific shRNA. OGT was hardly detected in the anti-SIRT1 co-immunoprecipitates from the SIRT1-silenced cells (Fig. 1a). We also performed a reciprocal co-immunoprecipitation (CoIP) assay. As shown in Fig. 1b, endogenous SIRT1 was readily co-immunoprecipitated with the OGT-specific antibody, and OGT-silencing abolished precipitation of SIRT1. Then, we determined the subcellular distribution of endogenous OGT and SIRT1 in NCI-H1299 cells to further

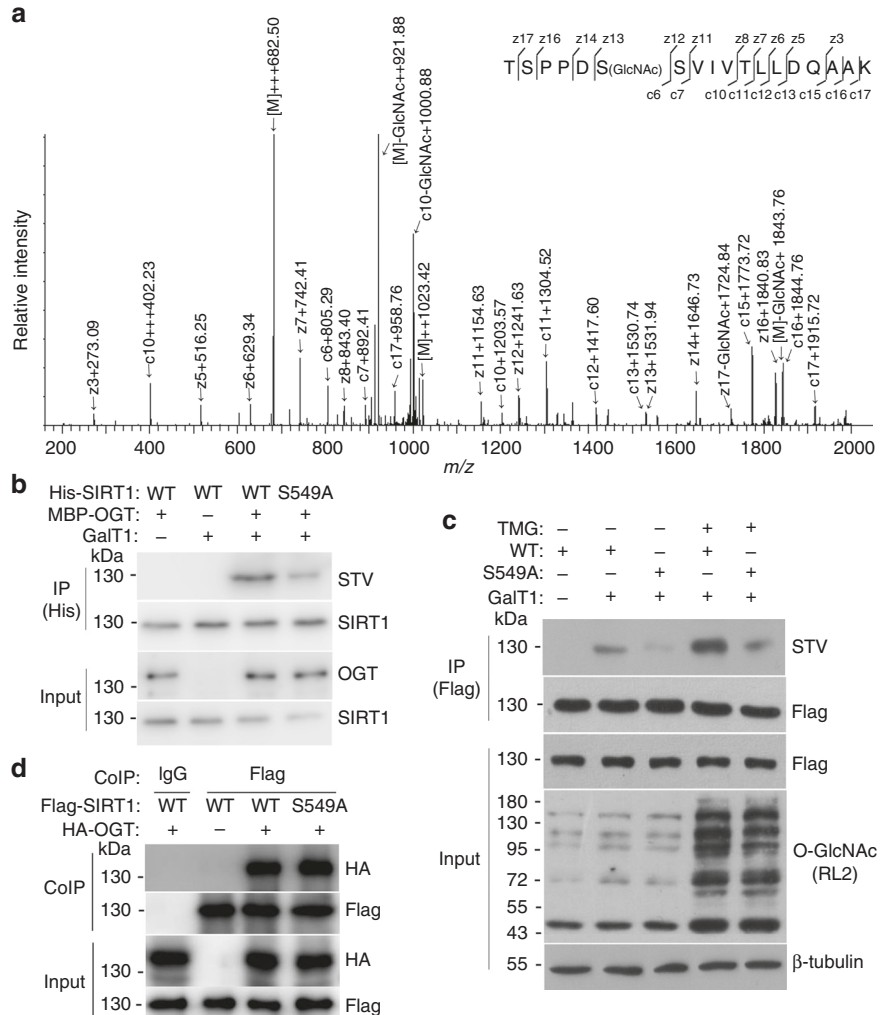

**Fig. 3** SIRT1 is O-GlcNAcylated at Ser549. **a** ETD-MS detected the O-GlcNAcylated peptide of SIRT1 (544–561). **b** Either His-tagged wtSIRT1 or SIRT1[S549A] was coexpressed with or without MBP tagged OGT in *E. coli*. SIRT1 was immunoprecipitated with anti-His antibody, and then O-GlcNAcylation of SIRT1 was detected by chemoenzymatic labeling and IB with STV-HRP. Data represent three independent experiments. **c** Flag-tagged wtSIRT1 or SIRT1[S549A] vector was transfected into NCI-H1299 cells. The cells were lysed after treatment with 2 μM TMG for 4 h, followed by chemoenzymatic labeling and IB. Data represent two independent experiments. **d** HA-tagged OGT and Flag-tagged wtSIRT1 or SIRT1[S549A] were co-transfected into NCI-H1299 cells, and the CoIP assays were carried out by anti-Flag antibody, followed by IB with anti-HA and anti-Flag antibodies. Data represent two independent experiments

substantiate their interaction. Immunofluorescence assays revealed that the majority of SIRT1 and OGT colocalized to the nucleus (Fig. 1c).

Next, we tested whether OGT directly binds SIRT1 using GST pull-down assays, and observed that recombinant SIRT1 interacts with GST-fused OGT but not GST in vitro (Fig. 1d). To identify the regions of SIRT1 that are responsible for the SIRT1–OGT interaction, we generated deletion mutants of SIRT1 fused with N-terminal GST tag. GST pull-down assays indicated that OGT bound to the C-terminal domain (aa 500–747; Fig. 1e, lane 4) but showed no affinity for either the amino (N)-terminal (Fig. 1e, lane 2) or central core domain of SIRT1 (Fig. 1e, lane 3).

**SIRT1 is modified by O-GlcNAc in vitro and in vivo.** The above results demonstrate that OGT binds SIRT1 both in vivo and in vitro, suggesting that SIRT1 may be a novel OGT substrate. An in vitro O-GlcNAcylation assay indicated that the recombinant SIRT1 could be O-GlcNAcylated by recombinant OGT (Fig. 2a). Therefore, GlcNAz metabolic labeling[36] and chemoenzymatic labeling[37], which are highly sensitive and unbiased methods, were used to detect O-GlcNAcylation of SIRT1 in vivo.

For metabolic labeling, the cultured cells were incubated with the azide-modified glucosamine (Ac4GlcNAz), and GlcNAz was incorporated into intracellular GlcNAc-containing glycoproteins. The cells were lysed by 1% dodecyl sulfate (SDS) lysis buffer and then the GlcNAz on the proteins were biotinylated with biotin alkyne via azide-alkyne cycloaddition (click chemistry). The SIRT1 proteins were immunoprecipitated and detected by probing with streptavidin-HRP (STV-HRP). The results indicated that SIRT1 in NCI-H1299 cells could be labeled by GlcNAz, and that the modification of SIRT1 was enhanced by overexpression of OGT (Fig. 2b).

For chemoenzymatic labeling, the cultured cells were lysed by 1% SDS lysis buffer and then diluted 10-fold with NP40 lysis buffer, which could denature proteins and disrupt protein complexes. As expected, SIRT1 was immunoprecipitated with antibody against SIRT1 and two SIRT1-associated proteins, SIRT1 negative regulator DBC1 and OGT, could not be detected in the precipitates (Supplementary Fig. 1). The precipitates were subsequently labeled with GalNAz using a mutant galactosyl-transferase (GalT1 Y289L) with an azide derivative of UDP-GalNAc (UDP-GalNAz) as donor substrate, followed by

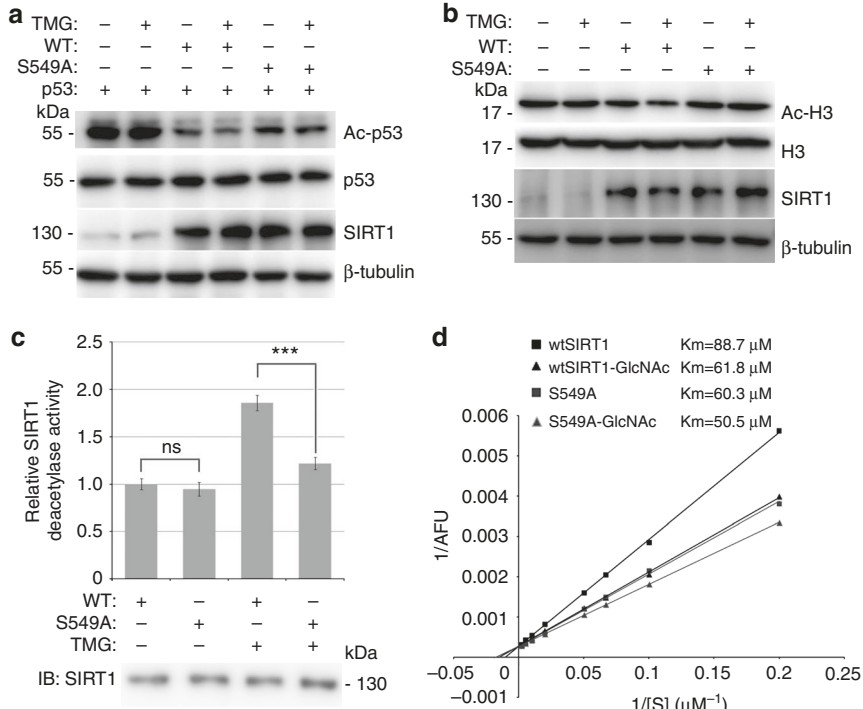

**Fig. 4** O-GlcNAcylation of SIRT1 at Ser549 increases its deacetylase activity. **a** NCI-H1299 cells were cotransfected with p53, shSIRT1 and either Flag-tagged wtSIRT1 or SIRT1[S549A] vectors. The cells were then treated by 1 μM TSA combining with or without 2 μM TMG for 3 h. The whole-cell extracts were immunoblotted with antibodies against p53 acetylated at Lys 382, p53, or SIRT1. Data represent three independent experiments. **b** NCI-H1299 cells were cotransfected with shSIRT1 and either Flag-tagged wtSIRT1 or SIRT1[S549A] vectors. The cells were then treated by 1 μM TSA combining with or without 2 μM TMG for 3 h. The whole-cell extracts were immunoblotted with antibodies against acetylated H3 (Lys 9/14), H3, or SIRT1. Data represent two independent experiments. **c** NCI-H1299 cells transfected with Flag-tagged wtSIRT1 or SIRT1[S549A] were treated with or without 2 μM TMG for 4 h. SIRT1 was immunopurified from cell extracts using anti-FLAG M2 Affinity Gel, and the relative deacetylase activity was determined in vitro at the concentration of 25 μM fluorogenic acetylated p53 peptide substrate and 500 μM NAD[+] using a fluorometric assay system. Data represent two independent experiments. **d** The enzyme kinetic parameters were detected by varying the p53 (K382Ac) fluorometric peptide substrate concentration while keeping NAD[+] at a saturating concentration for wtSIRT1 (black square), wtSIRT1 bearing O-GlcNAc (wtSIRT1-GlcNAc) (black triangle), SIRT1[S549A] (gray square), and SIRT1[S549A] bearing O-GlcNAc (SIRT1[S549A]-GlcNAc) (gray triangle). The Lineweaver–Burk plot and Km values were showed. Student's t-test. Results are expressed as the mean ± s.d. (n = 3 biologic replicates). P-values: ns (not significant), ***P < 0.001

biotinylating with biotin alkyne. Then, the labeled SIRT1 protein samples were detected by probing with STV-HRP. Figure 2c showed that SIRT1 was O-GlcNAcylated in various cell types, including HCT 116, NCI-H1299, PLC/PRF/5, and Hep G2 cells. The O-GlcNAcylation of transfected Flag-SIRT1 was also detectable by enzymatic labeling, which was obviously elevated by the treatment of the OGA specific inhibitor Thiamet-G (TMG) and overexpression of OGT (Fig. 2d). Additionally, an alternative experimental program for chemoenzymatic labeling was carried out to confirm SIRT1 O-GlcNAcylation. The NCI-H1299 cells were pretreated with TMG in order to elevate SIRT1 O-GlcNAcylation level. The cells were lysed by 1% SDS lysis buffer and directly labeled with GalNAz by GalT1, click on biotin, and then the biotinylated proteins were enriched using STV-conjugated magnetic beads. The well-known highly O-GlcNAcylated protein Nup62 was used as a positive control in this assay. The results showed that both Nup62 and SIRT1 proteins could be enriched from the cell lysate (Fig. 2e), and the results also suggested that only a small part of SIRT1 proteins were O-GlcNAcylated. Collectively, these results demonstrate that SIRT1 is a bona fide OGT substrate with dynamic O-GlcNAcylation in vivo.

**SIRT1 is O-GlcNAcylated at Ser 549**. O-GlcNAcylated SIRT1 was produced by co-expression of His-tagged SIRT1 and MBP-fused OGT in E. coli[38], after which the recombinant SIRT1

was purified by Ni-affinity chromatography and the O-GlcNAcylation of SIRT1 was detected (Supplementary Fig. 2a). To map the O-GlcNAc sites of SIRT1, the tryptic-digested peptides of SIRT1 were subjected to CID-based and ETD-based fragmentation using LC-MS/MS. In the CID-based ToF MS analysis, the O-GlcNAcylated 544–561 peptides of SIRT1 (TSPPDSSVIVTLLDQAAK) were observed at [M+3H][3+] m/z 682.50 (Supplementary Fig. 2b). Although the majority of the masses arise from the unmodified peptide, the spectrum showed the GlcNAc oxonium ion and several GlcNAc fragment ions. These indicated that the 544–561 peptide of SIRT1 carried O-GlcNAc, but the exact position of the O-GlcNAc site on this peptide was not identified because of the fragmentation of O-GlcNAc from the peptide during CID-MS analysis. To determine the exact O-GlcNAcylation site, ETD-MS analysis was performed. O-GlcNAcylation was mapped to Ser 549 (S549) within the 544–561 peptide region of SIRT1 (Fig. 3a). The S549 site is localized within the C terminal region, which is also responsible for OGT binding. A comparison of the S549 O-GlcNAcylation site of human SIRT1 with the same region from other mammals reveals that this position is highly conserved except in platypus (Supplementary Fig. 2c). These results suggest that SIRT1 O-GlcNAcylation may be highly conserved throughout mammalian evolution. However, this motif is not conserved among the seven members (SIRT1–7) of the human sirtuin family.

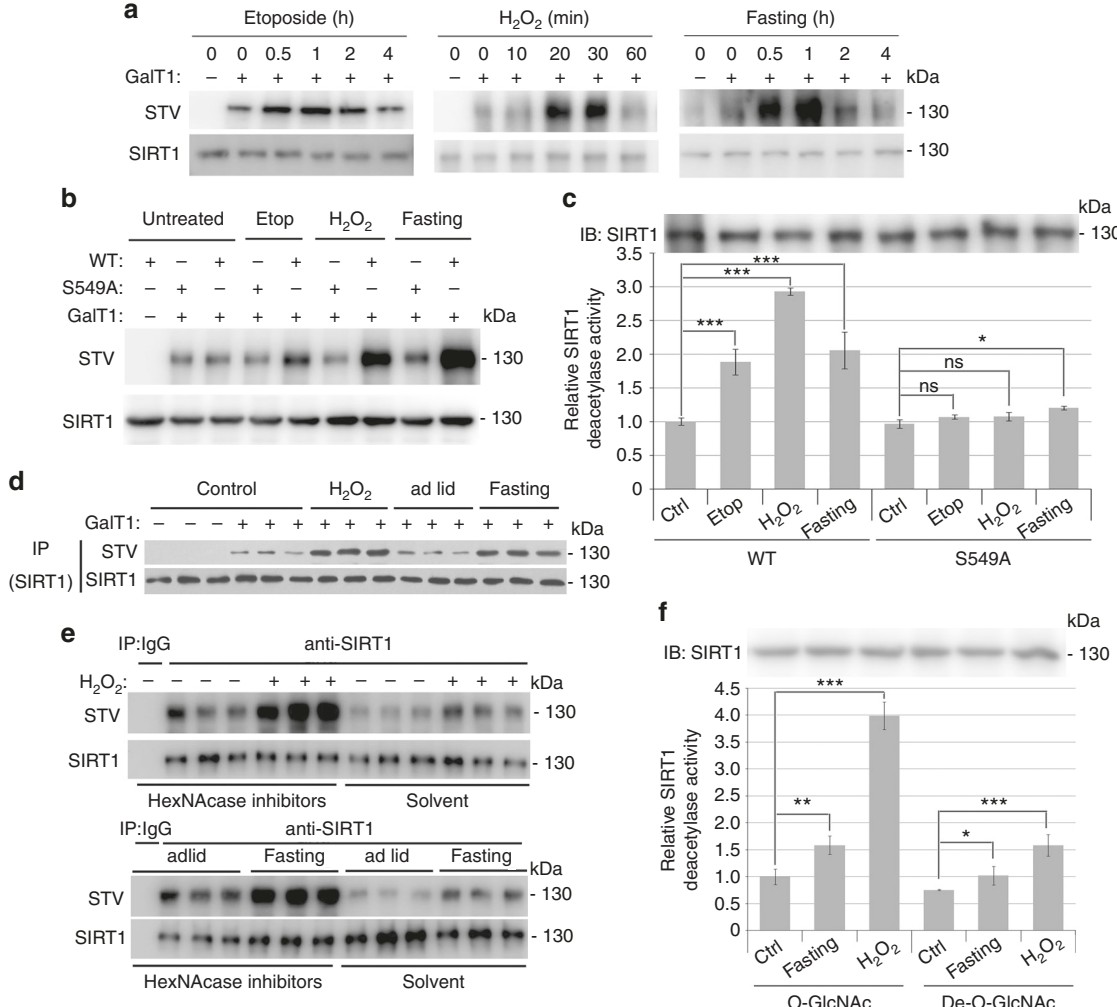

**Fig. 5** Stress stimuli induced O-GlcNAcylation and activation of SIRT1. **a** Stress stimuli induced SIRT1 O-GlcNAcylation in a time-dependent manner in NCI-H1299 cells (top). NCI-H1299 cells were stimulated with etoposide (25 μM), $H_2O_2$ (200 μM) or glucose depletion (fasting) for the indicated times and O-GlcNAcylation of SIRT1 was detected by chemoenzymatic labeling and IB analysis. Data represent two independent experiments. **b**, **c** NCI-H1299 cells were transfected with Flag-tagged wtSIRT1 or SIRT1^S549A, followed by treatment with etoposide (25 μM for 1 h), $H_2O_2$ (200 μM for 30 min) or glucose depletion (1 h). SIRT1 proteins were immunoprecipitated, and their O-GlcNAcylation levels (**b**) and deacetylase activities (**c**) were detected by chemoenzymatic labeling and fluorometric assay, respectively. Data represent two independent experiments. **d–f** Balb/c mice were either fasted for 18 h or injected intraperitoneally with 200 μl of 100 mM $H_2O_2$. SIRT1 proteins were immunoprecipitated from whole liver extracts. Chemoenzymatic labeling and IB analysis were used for the detection of SIRT1 O-GlcNAcylation levels (**d**). Mouse livers were homogenized in NP40 lysis buffer supplied with or without HexNAcase inhibitors, and SIRT1 proteins were immunoprecipitated from whole liver extracts. SIRT1 O-GlcNAcylation levels (**e**) and deacetylase activity (**f**) were detected by chemoenzymatic labeling and fluorometric assay system, respectively. For animal experiments, six mice per group were used, representative data from two independent experiments. For chemoenzymatic labeling, samples from two mice were mixed in equal amounts for detection. Student's t-test. Results are expressed as the mean ± s.d. (**c**, $n = 3$ biologic replicates; **f**, $n = 6$ mice per group). P-values: ns (not significant), *$P < 0.05$, **$P < 0.01$, ***$P < 0.001$

To determine whether S549 is a major site of O-GlcNAcylation, we coexpressed both wtSIRT1 and SIRT1 with an alanine substitution at S549 site (SIRT1^S549A) with OGT in *E. coli* and measured their relative levels of O-GlcNAcylation. The mutation of S549 to alanine led to a marked reduction in SIRT1 O-GlcNAcylation compared with wtSIRT1 (Fig. 3b). Then, Flag-tagged wtSIRT1 and SIRT1^S549A were exogenously expressed in NCI-H1299 cells, which were then treated with or without TMG before cell lysis, and the O-GlcNAcylation of SIRT1 was detected. The results indicated that mutating S549 to alanine visibly reduced SIRT1 O-GlcNAcylation in NCI-H1299 cells (Fig. 3c, lane 3). Meanwhile, TMG treatment markedly enhanced the O-GlcNAcylation of wtSIRT1 but not SIRT1^S549A (Fig. 3c, lanes 4 and 5), suggesting the O-GlcNAcylation at S549 is highly dynamic. The decrease in the O-GlcNAcylation of the SIRT1^S549A

mutant may also be due to the disruption of the interaction between OGT and SIRT1 by the S549A mutation. In order to rule out this possibility, the pull-down and CoIP assays were carried out. The results showed that both wtSIRT1 and SIRT1^S549A could effectively bind to OGT in vivo (Fig. 3d) and in vitro (Supplementary Fig. 3). Altogether, these results indicate that S549 is a major site for SIRT1 O-GlcNAcylation both in vitro and in vivo.

**O-GlcNAcylation at S549 enhances SIRT1 deacetylase activity.** To investigate the roles of O-GlcNAcylation on SIRT1 deacetylase activity, wtSIRT1 or SIRT1^S549A was expressed in NCI-H1299 cells and the SIRT1 activity was monitored by detecting the acetylation status of two SIRT1 targets, histone H3 and p53. In order to eliminate the role of endogenous SIRT1, NCI-H1299

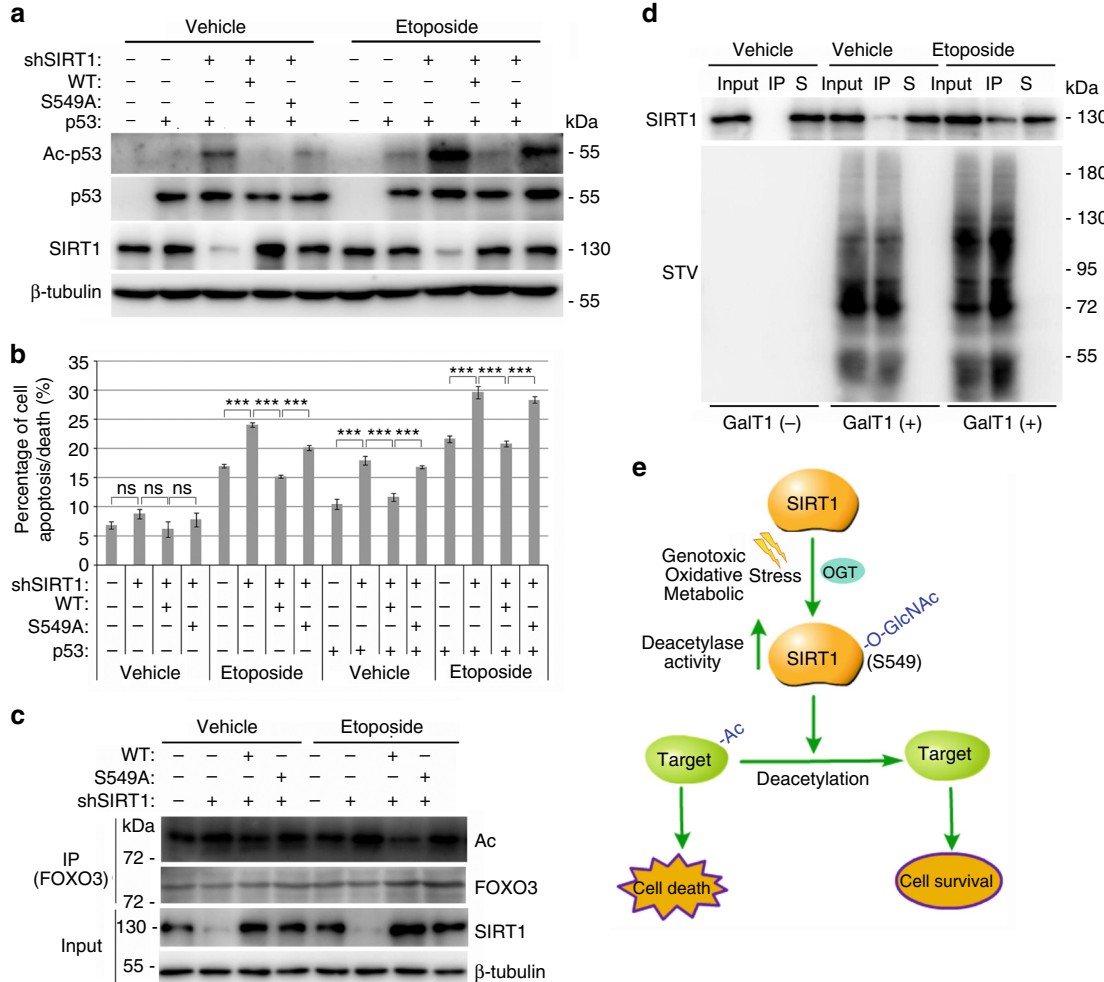

**Fig. 6** O-GlcNAcylation of SIRT1 protects cells from death during stress stimuli. **a** Flag-tagged wtSIRT1 or SIRT1$^{S549A}$ was co-expressed with or without p53 in NCI-H1299 cells, and then the cells were treated with 2 μM MG132, 1 μM TSA and either 25 μM etoposide or solvent for 2 h. The whole-cell extracts were immunoblotted with antibodies against p53 acetylated at Lys 382, p53, SIRT1 or β-tubulin. Data represent two independent experiments. **b** NCI-H1299 cells, which were stably silenced endogenous SIRT1 and reintroduced exogenous wtSIRT1 or SIRT1$^{S549A}$, were transfected with/without p53 and treated with etoposide. The percentage of cell apoptosis/death were analyzed by Muse Annexin V & Dead Cell Kit on Muse Cell Analyzer. Data represent three independent experiments. **c** NCI-H1299 cells, which were stably silenced endogenous SIRT1 and expressed wtSIRT1 or SIRT1$^{S549A}$, were treated with 2 μM MG132, 1 μM TSA and either 25 μM etoposide or solvent for 2 h. FOXO3 proteins were immunoprecipitated and detected by IB with antibodies against acetylation, FOXO3, SIRT1 or β-tubulin. Data represent two independent experiments. **d** NCI-H1299 cells, which were pretreated with or without 25 μM etoposide for 1 h, were lysed and labeled with GalNAz and biotin. The biotinylated proteins were enriched by STV-conjugated magnetic beads, and then the total protein samples (input), the immunoprecipitated protein samples (IP) and the remaining supernatant samples (S) were probed with indicated antibodies. Data represent two independent experiments. **e** Schematic representation of a model of how SIRT1 O-GlcNAcylation at S549 regulates its deacetylase activity and cellular response to stress stimuli. Student's *t*-test. Results are expressed as the mean ± s.d. (*n* = 3 biologic replicates). *P*-values: ns (not significant), ***P < 0.001

cells were first transfected with shRNA targeting the 3′-untranslated region (3′-UTR) of SIRT1 mRNA that selectively reduced endogenous, but not exogenous SIRT1 (Supplementary Fig. 4). TMG treatment obviously decreased the acetylation levels of exogenous p53 and endogenous H3 in wtSIRT1-transfected cells but not in the SIRT1$^{S549A}$-transfected cells (Fig. 4a, b), suggesting that O-GlcNAcylation at S549 is a positive regulator of SIRT1 deacetylase activity in vivo. Additionally, the deacetylase activities of immunopurified Flag-wtSIRT1 and Flag-SIRT1$^{S549A}$ were measured using an in vitro fluorometric assay with a p53 peptide acetylated at K382 as the substrate. The results showed that there was no significant difference in the deacetylase activity between wtSIRT1 and SIRT1$^{S549A}$ under basal conditions, but TMG treatment enhanced the activity of wtSIRT1 but not SIRT1$^{S549A}$ (Fig. 4c).

In order to understand the mechanism by which O-GlcNAcylation enhances SIRT1 deacetylase activity, enzyme kinetic parameters using an in vitro fluorometric assay with a p53 peptide acetylated at K382 as the substrate were determined. The wtSIRT1 and SIRT1$^{S549A}$ were co-expressed with/without OGT in *E. coli*, and then recombinant SIRT1 proteins were purified. SIRT1 proteins and their O-GlcNAcylation status were examined (Supplementary Fig. 5a). The enzyme kinetic assays showed that O-GlcNAcylation obviously enhanced substrate affinity (Km) and catalytic efficiency (kcat/Km) of wtSIRT1 for acetylated p53 peptide (Fig. 4d and Supplementary Table 1). However, the effects of O-GlcNAcylation on SIRT1$^{S549A}$ were not as obvious as that on wtSIRT1 (Fig. 4d and Supplementary Table 1). Consistent with the above results, SIRT1 deacetylase activity assays indicated that mutation of S549 to alanine slightly enhanced SIRT1

deacetylase activity. However, when coexpressed with OGT, SIRT1 deacetylase activity of SIRT1[S549A] was markedly decreased compared with wtSIRT1 (Supplementary Fig. 5b). These results demonstrate that O-GlcNAcylation at S549 is important for the regulation of catalytic efficiency and substrate affinity of SIRT1.

To determine whether O-GlcNAcylation of SIRT1 at S549 site could affect its subcellular localization, the GFP fusion proteins with either wtSIRT1 or SIRT1[S549A] were exogenously expressed in NCI-H1299 cells. The results showed that both wtSIRT1 and SIRT1[S549A] were predominantly detected in the nucleus and TMG treatment did not affect the localization of SIRT1 (Supplementary Fig. 6a). Immunofluorescence assays were also performed using anti-Flag antibody to detect transiently trans-fected Flag-SIRT1 and Flag-SIRT1[S549A] in NCI-H1299 cells, which supported the above results (Supplementary Fig. 6b).

**Stress stimuli activate SIRT1 by O-GlcNAcylation.** Considering the previous reports that both O-GlcNAc and SIRT1 serve as stress sensors, as well as our findings that O-GlcNAcylation of SIRT1 increases its activity, we hypothesized that O-GlcNAcylation of SIRT1 might be involved in the cellular stress response. To test this hypothesis, O-GlcNAcylation of SIRT1 was detected under stress-stimulated conditions in both cellular and mouse models. The results showed that both apoptotic agents (etoposide and hydrogen peroxide) and fasting (glucose deprivation) induced the O-GlcNAcylation of endogenous SIRT1 in NCI-H1299 cells in a highly dynamic time-dependent manner (Fig. 5a). We also revealed that stress stimuli elevated the O-GlcNAcylation of wtSIRT1 more so than that of SIRT1[S549A] (Fig. 5b). Similar results were obtained using HCT 116 cells (Supplementary Fig. 7). To investigate whether the elevation of SIRT1 O-GlcNAcylation was responsible for the activation of SIRT1 upon encountering stress stimuli, exogenously expressed Flag-wtSIRT1 and Flag-SIRT1[S549A] were immunopurified from NCI-H1299 cells and their deacetylase activities were assayed. Figure 5c showed that stress stimuli enhanced the activity of wtSIRT1 but not SIRT1[S549A]. These results indicate that O-GlcNAcylation is a dynamic positive regulator for the activation of SIRT1 at the early stage of stress response.

To investigate the potential roles of O-GlcNAc in the regulation of SIRT1 after exposure to stress stimuli in mouse model, Balb/c mice were either fasted or injected intraperitoneally with $H_2O_2$. IB and real-time PCR (RT–PCR) assays were performed to detect molecular markers of stress responses in mouse liver (Supplementary Fig. 8). Consistent with the cellular results, both $H_2O_2$ and fasting markedly enhanced SIRT1 O-GlcNAcylation in mouse livers (Fig. 5d). To evaluate whether the elevation of SIRT1 O-GlcNAcylation contributed to SIRT1 activation after stress stimuli in mouse liver, we obtained SIRT1 proteins retaining O-GlcNAcylation as well as de-O-GlcNAcylated SIRT1 proteins from the mouse livers and performed deacetylase activity assays in vitro. O-GlcNAcylation is highly labile in cell lysates due to the presence of β-N-acetylhexosaminidases (HexNAcases), thus HexNAcase inhibitor cocktails were added to the lysis buffer during the SIRT1 extraction and purification processes to maintain its O-GlcNAcylation status. On the other hand, buffers without any HexNAcase inhibitor were used to immunopurify de-O-GlcNAcylated SIRT1. The detection of SIRT1 O-GlcNAcylation indicated that we obtained both O-GlcNAcylated and de-O-GlcNAcylated SIRT1 proteins (Fig. 5e). The deacetylase activity assays showed that both fasting and intraperitoneal $H_2O_2$ injection significantly increased SIRT1 deacetylase activity whereas de-O-GlcNAcylation almost restored the activity to basal levels (Fig. 5f), indicating that O-GlcNAcylation of SIRT1

directly enhances its intrinsic deacetylase activity in mouse liver after exposure to metabolic and oxidative stresses.

**SIRT1 O-GlcNAcylation promotes cell survival under stress.** SIRT1 has been characterized as promoting cytoprotection in response to stressful conditions[21], with one of the main mechanisms through the deacetylation of p53[7, 39]. Based on the above findings, we proposed that O-GlcNAcylation of SIRT1 might play important roles in response to stress stimuli by deacetylating p53. To elucidate this notion, we examined whether O-GlcNAcylation of SIRT1 could decrease the acetylation level of p53 induced by etoposide. The proteasome inhibitor MG132 was added to cell cultures prior to introducing stress stimuli, which could normalize the expression of p53. The results showed that etoposide treatment elevated the K382 acetylation of exogenously expressed p53 in NCI-H1299 cells (Fig. 6a). Although the expression of either wtSIRT1 or SIRT1[S549A] could decrease p53 acetylation, the deacetylase activity of wtSIRT1 was much stronger (Fig. 6a).

To investigate the functional consequences of p53 deacetylation induced by SIRT1 O-GlcNAcylation, we examined the effects of SIRT1 O-GlcNAcylation on p53-mediated transcriptional activation using a p53-driven luciferase reporter construct. As expected, the expression of either wtSIRT1 or SIRT1[S549A] repressed p53-activated luciferase reporter gene transcription under both basal and stress conditions; however, the repressive ability of wtSIRT1 was much stronger than that of SIRT1[S549A] under stressful conditions (Supplementary Fig. 9a). Then, the transcription of the endogenous p53-target gene Cdkn1a was detected by RT–PCR. The results showed that wtSIRT1 expression attenuated p53-induced Cdkn1a gene transcription in both the basal and etoposide treatment conditions, but the effect of SIRT1[S549A] was less pronounced (Supplementary Fig. 9b).

These above results indicated that O-GlcNAcylation of SIRT1 attenuated the function of p53 during the stress response, suggesting that SIRT1 O-GlcNAcylation might prevent stress-induced apoptosis. To test this hypothesis, NCI-H1299 cells were stably infected with lentiviruses expressing shRNA targeting the 3′-UTR of SIRT1 combined with retroviruses expressing wtSIRT1 or SIRT1[S549A] (Supplementary Fig. 10). The cells were then transfected with/without p53 and treated with etoposide. The death and apoptosis of cells were analyzed. As expected, both etoposide treatment and exogenous p53 expression induced cell death, and SIRT1 silenced cells were more susceptible to death (Fig. 6b and Supplementary Fig. 11). More importantly, the results showed that the expression of wtSIRT1 obviously protected cells from death, but the effects of SIRT1[S549A] were very weak (Fig. 6b). Then, the presence of activated caspase-3 and cleaved PARP was also detected by IB analysis (Supplementary Fig. 12a), which supported the above results. The similar results were obtained by detecting the activity of caspase3/7 (Supplementary Fig. 12b). O-GlcNAcylation of SIRT1 also played cytoprotective role in HCT 116 cells (Supplementary Fig. 13).

Although the above results indicated that O-GlcNAcylation of SIRT1 protected cells from death by deacetylating and inactivating p53, the evidences also showed that O-GlcNAcylation of SIRT1 could protect cells in the absence of p53. Considering that SIRT1 can deacetylate a wide range of substrate proteins, some of them, such as FOXO3[8], can also mediate the cytoprotective role of SIRT1, the acetylation status of FOXO3 was detected. The results showed that wtSIRT1, but not SIRT1[S549A], obviously deacetylated FOXO3 in NCI-H1299 cells (Fig. 6c).

As a highly dynamic PTM, the O-GlcNAcylation is often substoichiometric on a protein. In order to better understand the regulatory role of SIRT1 O-GlcNAcylation in cytoprotection, the

percentage of the O-GlcNAcylated SIRT1 population in NCI-H1299 stimulated with or without etoposide was assessed. To do this, the cells were lysed, labeled with GalNAz, click on biotin, and then the biotinylated proteins were pulled down using STV-conjugated magnetic beads. The results showed that almost all of the biotinylated proteins were enriched from the samples, and much more SIRT1 proteins were pulled down from etoposide pre-treated cell sample (Fig. 6d). At the same time, a dilution series of cell lysate were immunoblotted with anti-SIRT1 antibody and a standard curve was generated (Supplementary Fig. 14). Based on this standard curve, the percentage of O-GlcNAcylated SIRT1 was about 4% under basal condition, but increased to about 12% after etoposide treatment, suggesting that O-GlcNAcylation of SIRT1 only participated in a small number of SIRT1's functional complexes. However, we might under-estimate the level of SIRT1 O-GlcNAcylation because O-GlcNAcylation is very fragile and can be easily lost during the course of the experiment.

These findings suggest that O-GlcNAcylation of SIRT1 at S549 is a molecular switch that regulates the balance between survival and death during the cellular stress response via deacetylating certain proteins, such as p53 and FOXO3 (Fig. 6e).

## Discussion

As SIRT1 is an important guardian of mammalian healthspan, precise regulation of its function is essential. In this study, we discovered that SIRT1 was O-GlcNAcylated. O-GlcNAcylation of SIRT1 at the S549 site directly enhances the deacetylase activity of SIRT1. Upon exposure to stress stimuli, the O-GlcNAcylation levels and consequent activity of SIRT1 are substantially elevated and protect cells from stress-induced death/apoptosis.

O-GlcNAc signaling is known to be an essential stress and metabolic sensor, and global O-GlcNAcylation levels and OGT activity are modulated by various stress stimuli[35, 40, 41]. In almost all of the mammalian cell/tissue types examined so far, one of the earliest responses to cellular stress is the elevation of O-GlcNAcylation on many proteins[35]. Our studies demonstrate that SIRT1 is a novel target of O-GlcNAc, and O-GlcNAcylation of SIRT1 activates its deacetylase activity, which also endows SIRT1 with a rapid stress response. Apart from O-GlcNAc, SIRT1 is also modified and modulated by other types of PTMs, including phosphorylation[27, 28, 42–46], methylation[47], nitrosylation[48], and SUMOylation[26]. These findings raise the possibility that SIRT1 activity may be controlled through the complex, combinatorial effects of various PTMs depending on the type, strength and temporal influence of stress stimuli, as well as tissue types.

In this study, we found S549 is a major and critical O-GlcNAcylation site on SIRT1. The O-GlcNAcylation at S549 was mapped by analyzing the in vitro O-GlcNAcylated SIRT1. SIRT1 O-GlcNAcylation was markedly reduced, although not completely eliminated, by the mutation of S549 to alanine. And the mutation reduced SIRT1 O-GlcNAcylation more obviously when the culture cells were stimulated by different types of stress or treated with OGA inhibitor. These results indicated that S549 is a major and dynamic O-GlcNAcylation site on SIRT1. In terms of biological function, O-GlcNAcylation obviously enhances the deacetylase activity of cellular wtSIRT1, but not SIRT1$^{S549A}$. However, there were some contradicting results when the SIRT1 proteins modified by OGT in vitro were examined. In addition to O-GlcNAcylation can obviously promote the deacetylase activity of wtSIRT1 (about 3-fold), there is still an increase in deacetylase activity between O-GlcNAcylated SIRT1$^{S549A}$ (1.8-fold) and unmodified SIRT1$^{S549A}$ (1.4-fold) (Supplementary Fig. 5b), sug-gesting that there are additional sites of O-GlcNAcylation on SIRT1 involved in its deacetylase activity. Consistent with the

above contradiction, there are sites modified by OGT in vitro that are not modified in vivo at high stoichiometry, as evidenced by the ability to detect in vitro, but not in vivo, O-GlcNAcylated SIRT1 with RL2 antibody. One possible explanation is that the binding of some protein partners to SIRT1 hinders the mod-ification of these sites by OGT in cells. This may also be due to the high level of OGT expression in E. coli. Collectively, these results demonstrated that S549 is a major and critical O-GlcNAcylation site on SIRT1 in cells.

The health of an organism is regulated by multiple molecular and biochemical networks responsible for sustaining homeostasis within cells and tissues. However, the homeostatic imbalance often occurred in response to a variety of endogenous and environmental stresses, allowing for the accumulation of damage and increased susceptibility to diseases[49]. SIRT1 plays a vital role in stress management and cytoprotection[21]. Our results indicate that genotoxic, oxidative, and metabolic stress stimuli enhance the O-GlcNAcylation and activity of SIRT1. All of these types of stress stimuli are involved in SIRT1-regulated physiological and pathological events such as chronic inflammatory diseases and metabolic dysfunctions, including ageing, obesity, diabetes, and cancer[50]. Additionally, O-GlcNAc could modify and modulate SIRT1 in both human and mouse, which may also exist through mammalian SIRT1 because of the highly conserved S549 site. Collectively, these findings suggested that O-GlcNAcylation of SIRT1 may be an evolutionarily conserved process to drive cel-lular homeostasis by invoking a variety of stress response pathways.

Although the relevance of SIRT1 as a longevity gene has been disputed, it is evident that SIRT1 impacts healthspan by reg-ulating the activity of its target enzymes and transcription factors to drive the cell toward cytoprotection in response to stress[48]. Small molecule SIRT1 activators are theoretically capable of delaying ageing and preventing age-related diseases[51]. Thus, our finding that O-GlcNAc increases SIRT1 deacetylase activity is clinically relevant, and agents that regulate SIRT1 O-GlcNAcylation are potential novel drugs for preventing age-related diseases and extending healthspan.

## Methods

**Cell culture**. The cell lines HEK 293T, NCI-H1299, HCT 116, Hep G2, and PLC/PRF/5 were kindly provided by Stem Cell Bank, Chinese Academy of Sciences. All the cells have been authenticated by short tandem repeats analysis and tested for mycoplasma contamination by PCR. HEK 293T, HCT 116, Hep G2, and PLC/PRF/5 cells were cultured in Dulbecco's Modified Eagle Medium (HyClone, SH30243.01B) with 10% fetal bovine serum (FBS; Invitrogen, 10099-141). NCI-H1299 cells were cultured in RPMI-1640 (HyClone, SH30809.01) medium sup-plemented with 10% FBS.

**Plasmids**. The pcDNA3.1-Myc/His-SIRT1 plasmid was obtained from Dr. Tony Kouzarides (University of Cambridge, Cambridge, UK)[17]. The coding region of wtSIRT1 with a Myc-tag and a His-tag at the 3′ terminal was cloned into pET24a (+) plasmid. The coding region of wtSIRT1 with a 3×Flag-tag coding sequence at the 3′ terminal was cloned into MSCVpuro and pEGFP-N2 plasmids. The S549A SIRT1 mutant was generated using the QuikChange site-directed mutagenesis kit (Stratagene). The puromycin resistant pLKO.1 vector was used to construct the SIRT1 shRNA vector, the targeting sequence is GAAGTGCCTCAGATATTAA. The hygromycin B resistant pLKO.1 vector was used to construct the SIRT1 shRNA vector targeting the 3′-UTR of SIRT1, the targeting sequence is GCTAAGAATTTCAGGATTA. The GST fused full-length SIRT1 and deleted mutant SIRT1 expression vectors were constructed based on pGEX-2T vector (GE Healthcare). The p3×Flag-OGT plasmid was provided by Dr. Cho (Yeonsei Uni-versity)[52]. The pDEST-HA-OGT plasmid was provided by Dr. G.W. Hart (Johns Hopkins University). The pMal-c2×-OGT recombinant expression vector was obtained from Dr. D.J. Vocadlo (Simon Fraser University)[38]. The pGEX-2T-OGT was provided by Suat Özbek (University of Heidelberg)[53]. The p53 luciferase reporter construct p53-TA-luc containing p53 binding sites driving firefly luci-ferase expression from a minimal herpes simplex virus thymidine kinase promoter was obtained from Clontech.

**Transient transfection and generation of stable cell lines.** Transient transfection of HEK 293T cells was performed using Attractene Transfection reagents (QIAGEN), according to the manufacturer's instructions. For other cell lines, transient transfection was performed by electric transfection using X-Porator EBXP-H1 (ETTA). Stable cell lines were obtained by retroviral or lentiviral infection and selection with puromycin (1 µg ml$^{-1}$) or hygromycin B (200 µg ml$^{-1}$) for 2 weeks.

**Immunoprecipitation (IP) and CoIP.** For IP assay, cells were lysed in RIPA buffer (25 mM Tris-HCl [pH 7.4], 150 mM NaCl, 1% NP40, 0.1% SDS, 1 mM EDTA, 1 mM Na3VO4, 10 mM NaF, 10 µM PUGNAc [Sigma], 2 mM STZ [Sigma], 10 µM TMG [Sigma], 40 mM GlcNAc [Sigma], and Complete™ protease inhibitors [Roche]). For CoIP assay, cells were lysed in NP40 lysis buffer (25 mM Tris-HCl [pH 7.4], 150 mM NaCl, 1% NP40, 1 mM EDTA, 1 mM Na3VO4, 10 mM NaF, 10 µM PUGNAc, 2 mM STZ, 10 µM TMG, 40 mM GlcNAc, and Complete protease inhibitors).

**Immunoblotting.** Cells were lysed in SDS lysis buffer (1% SDS, 50 mM Tris-HCl, pH 7.5, 100 mM NaCl, 2 mM STZ, 40 mM GlcNAc, and Complete™ protease inhibitors), and the lysate was resolved by 4–12% SDS polyacrylamide gels (SDS-PAGE), transferred to Immobilon-FL PVDF membrane (#IPVH00010, Merck Millipore), and immunoblotted with the indicated antibodies. Blots were probed with antibodies recognizing, dilutions and clone/catalog numbers in brackets: SIRT1 (1:1000, #9475, Cell Signaling Technology; 1:1000, #07-131, Upstate), p53 (1:1000, #2524, Cell Signaling Technology), Acetyl-p53 (Lys382) (1:1000, #2525, Cell Signaling Technology), OGT (1:1000, #ab96718, Abcam), O-GlcNAc (RL2, 1:1000, #ab93858, Abcam; CTD110.6, 1:1000, #MMS-248R, Covance), GST-Tag (1:2000, #ab9085, Abcam), Flag-Tag (1:2000, #F7425, Sigma-Aldrich), His-Tag (1:2000, #M20001, Abmart), Myc-Tag (1:2000, #M20002, Abmart), HA-Tag (1:1000, #M20003, Abmart), histone H3 (1:1000, #CST9715, Cell Signaling Technology), Acetyl-H3 (Lys9/14) (1:200, #sc-8655, Santa Cruz), FOXO1 (1:1000, #CST2880, Cell Signaling Technology), Acetyl-FOXO1 (1:200, #sc-49437, Santa Cruz), FOXO3 (1:2000, #ab12162, Abcam), Acetylated-Lysine (1:1000, #CST9441, Cell Signaling Technology), p21 (1:1000, #ab109199, Abcam), DBC1 (1:1000, #ab70239, Abcam), activated caspases-3 (1:1000, #CST9661, Cell Signaling Technology), cleaved PARP (1:1000, #CST5625, Cell Signaling Technology), β-tubulin (1:2000, #M20005, Abmart), GAPDH (1:5000, #M20006, Abmart). Uncropped western blot gels are shown in Supplementary Figs. 15–20.

**Immunofluorescence microscopy.** Cells were fixed in 4% (wt/vol) paraformaldehyde for 15 min, washed with PBS and permeabilized with PBS containing 0.5% (vol/vol) Triton X-100 for 3 min. Cells were then washed with PBS, incubated with the indicated primary antibodies (anti-Flag antibody, #F7425; anti-SIRT1 antibody, #ab110304; anti-OGT antibody, #ab96718) overnight at 4 °C, washed with PBS and incubated for 45 min with the appropriate secondary antibodies (Alexa Fluor 488-conjugated goat anti-mouse IgG (H+L) secondary antibody, #A-11029; Alexa Fluor 594-conjugated goat anti-rabbit IgG (H+L) secondary antibody, #A-11012) at room temperature. After washing, coverslips were mounted using ProLong Gold mounting medium (Invitrogen). Images were acquired on either an Olympus FV1000 confocal microscope with a silicon immersion 60× objective or a Zeiss LSM 510 Meta with a silicon immersion 40× objective.

**O-GlcNAc enzymatic labeling.** Cells were lysed in RIPA buffer. Either endogenous SIRT1 or exogenously expressed Flag-SIRT1 proteins were immunoprecipitated from cell lysate (800 µg) using either anti-SIRT1 antibody (1:100, #07-131) or anti-FLAG M2 Affinity Gel (Sigma). The immunopurified SIRT1 was enzymatically labeled with an azido-containing nucleotide sugar analog (UDP-GalNAz) using an engineered (1,4)-galactosyltransferase (GalT1 Y289L) according to the Click-iT O-GlcNAc enzymatic labeling kit protocol (Invitrogen) and conjugated with an alkyne-biotin compound as per the Click-iT protein analysis detection kit protocol (Invitrogen). Biotin-labeled samples were subsequently probed with STV-HRP. Control experiments were performed in parallel in the absence of GalT1 Y289L.

**Protein purification.** All the recombinant proteins (His-SIRT1, GST-SIRT1, MBP-OGT, and GST-OGT) were expressed in *E. coli* BL21 (DE3) cells. The BL21 (DE3) cells harboring the indicated protein expression constructs were grown at 37 °C in LB medium and induced at OD600 = 0.4–0.6 by 0.1 mM Isopropyl β-D-thiogalactopyranoside at 20 °C overnight. The cells were harvested and lysed by ultra high-pressure homogenization in a lysis buffer containing 50 mM Tris (pH 7.5), 150 mM NaCl with the addition of 0.1 mg ml$^{-1}$ PMSF. For purification of O-GlcNAcylated SIRT1, 10 µM PUGNAc, 2 mM STZ, 10 µM TMG, and 40 mM GlcNAc were added to the buffer. The cell debris was removed from lysates by centrifugation. For the purification of His-tagged proteins, the protein supernatant was loaded on to the HisTrap HP column (GE Healthcare, #17-5247-01) and eluted by lysis buffer containing 200 mM imidazole (Sigma). For the purification of GST-fused proteins, the protein supernatant was loaded on to the GSTrap FF column (GE Healthcare, #17-5130-02) and eluted by lysis buffer containing 20 mM reduced glutathione (Sigma). For the purification of MBP-fused proteins, the protein

supernatant was loaded on to the MBPTrap HP column (GE Healthcare, #29-0486-41) and eluted by lysis buffer containing 10 mM maltose (Sigma). For purification of the His-SIRT1 bearing O-GlcNAc, the His-tagged SIRT1 and MBP-fused OGT were co-expressed in BL21(DE3) strain, and recombinant SIRT1 was purified by HisTrap HP column, and then the eluted protein samples were loaded on to the MBPTrap HP column for removing the co-purified MBP-OGT. All the purified proteins were desalted and concentrated by ultrafiltration using Centrifugal Filters Ultracel-50K (Millipore, #UFC905096).

**In vitro O-GlcNAcylation assay.** In vitro O-GlcNAcylation of SIRT1 (500 ng, binding on Ni-NTA Agarose beads) was performed in 100 µl assay volumes containing 500 ng of OGT, in reaction buffer (50 mM Tris-HCl, pH 7.5, 1 mM DTT, 12.5 mM MgCl$_2$), and 1 mM UDP-GlcNAc. The reactions were incubated for 3 h at 37 °C, and then the beads were washed three times with NP40 lysis buffer. The bound proteins were eluted with SDS sample buffer before electrophoresis using SDS-PAGE.

**O-GlcNAc site mapping of SIRT1.** The His-tagged SIRT1 and MBP-fused OGT were co-expressed in *E. coli* BL21(DE3) strain, and recombinant SIRT1 was purified by Ni-affinity chromatography. After electrophoresis using SDS-PAGE and staining with Bio-Safe Coomassie blue R250 Stain (Bio-Rad), the SIRT1 protein band was excised and manually digested in the gel with chymotrypsin (Promega)[54]. Sample fractions were analyzed by LC-MS/MS. Chromatography was performed using a LC-20AD nano HPLC (Shimadzu) at a flow rate of 400 nL/min. The column was a BEH C18 Column 75 µM × 100 mm (Waters). Peptides were eluted by a gradient from 2 to 35% solvent B (98% ACN, 0.1% formic acid) for 44 min followed by a short wash at 80% solvent B and then solvent A (2% ACN, 0.1% formic acid), before returning to starting conditions. Then the eluted peptide components were analyzed by using either an LTQ Orbitrap XL™ ETD mass spectrometer (Thermo Scientific) equipped with an ETD option or a TripleTOF 5600 (AB SCIEX, Concord, ON) equipped with Nanospray III source. For electrospray ionization (ESI) MS/MS analysis performed on the TripleTOF 5600, the nanospray voltage was typically 2500 V in the nano-LC (liquid chromatography) ESI MS/MS mode. For LTQ Orbitrap XL™ ETD mass spectrometer analysis, the ion source voltage is set to 1500 V. MS/MS data were searched using the pFind program for O-GlcNAcylated peptides, and the spectra were annotated using pLabel[55–57].

**GST pull-down assay.** The collected purified proteins (300 ng) were incubated with GST fusion proteins (100–200 ng) bound on GS4B beads (GE Healthcare) overnight at 4 °C. The beads were then washed three times with NP40 lysis buffer, and the bound proteins were eluted with SDS sample buffer before electrophoresis using SDS-PAGE.

**SIRT1 activity assay.** SIRT1 deacetylase activity was measured with a SIRT1 fluorimetric activity assay/drug discovery kit (catalog #AK555, Biomol International LP). For determination of the relative SIRT1 deacetylase activity in vitro, purified SIRT1 protein was incubated with 25 µM fluorogenic acetylated p53 peptide substrate and 500 µM NAD$^+$ (or the indicated concentrations of fluorogenic acetylated peptide substrate and NAD$^+$) for 20 min at 37 °C. For determination of enzyme kinetic parameters, the reaction contained a saturating concentration of NAD$^+$ (3 mM) while varying fluorogenic acetylated p53 peptide concentrations (0–500 µM). The reactions were incubated for 5 min at 37 °C. Reactions were halted by the addition of 1 mM nicotinamide, and the deacetylation-dependent fluorescent signal was determined using a 360-nm excitation laser and a 460-nm emission filter on a fluorescence plate reader. Background control reactions were performed in the absence of enzyme. All of the reactions were performed in triplicate. Km and kcat values were obtained by fitting the data to the Michaelis–Menten equation.

**Luciferase reporter assay.** For reporter assays, cells were transfected using electroporation with plasmids as described in the figure legend. At 48 h after transfection, the cells were lysed and the luciferase activities were measured using the Dual-Luciferase Reporter Assay System (Promega), according to the manufacturer's instructions. The relative luciferase activity was determined by normalization with Renilla luciferase activity.

**Caspase-3/7 activity assay.** Twenty-four hours after transfection, the cells were seeded in white-walled 96-well plates at the density of $5 \times 10^3$ cells in 100 µl and incubated for 24 h. Then, 1 µM TSA (Sigma) and either 25 µM etoposide (Sigma) or DMSO (solvent control) were added, and cells were incubated for an additional 6 h. The caspase-3/7 activity was measured using the Caspase-Glo 3/7 assay kit (Promega), according to the manufacturer's instruction. Luminescence was measured using an EnSpire multiwell plate reader (Perkin Elmer). Caspase-3/7 activity was expressed as fold change compared with the untreated control.

**Analysis of cell death**. The death and apoptosis of cells were detected by Muse$^{TM}$ Annexin V & Dead Cell Kit (Muse$^{TM}$ Cell Analyzer, Millipore), according to the manufacturer's protocol.

**Quantitative RT–PCR**. Detection of mRNA levels was performed using a 7500 Real-Time PCR System (Applied Biosystems) and SYBR Green master mix (Qiagen). Total RNA was extracted using TRIzol (Invitrogen) and first-strand cDNA was synthesized using M-MLV Reverse Transcriptase (Promega), according to the manufacturer's instructions. Primers that amplify stress response genes were designed with Primer5 software (Premier Biosoft International). PCR primers are listed in Supplementary Table 2. Quantitative RT–PCR was performed in triplicate, and the mRNA levels of target genes were normalized to GAPDH.

**Mice**. Female Balb/c mice (8–10-weeks-old) were purchased from the Vital River Laboratory (Charles River China) and raised in a specific pathogen-free and air-conditioned animal facility. Mice were fed ad libitum with a standard diet (Shanghai Laboratory Animal Company) and kept under a 12-h light-dark cycle. No randomization was performed. The studies were not conducted in a blinded manner. Fasting in mice was performed by providing the animals only water for 18 h, prior to the experiments. Oxidative stress treatment was performed by intra-peritoneal injection of 200 μl of 100 mM $H_2O_2$. Six mice per group were used and samples from two mice were mixed in equal amounts for detection. All animal studies were performed according to protocols reviewed and approved by the Ethics and Animal Welfare Committee of the School of Medicine and Pharmacy at Ocean University of China.

**Statistical analysis**. For comparison between two groups, normality was not assessed for the statistical analysis. Data were analyzed by the Student's $t$-test using the SPSS 11.0 software program (SPSS). Data are present as the mean ± standard deviation (s.d.). $P < 0.05$ was considered statistically significant.

**Data availability**. All data supporting the findings of this study are available within the article and its Supplementary Information files or from the corresponding authors on reasonable request.

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

## Acknowledgements

We thank Dr. T. Kouzarides (University of Cambridge, Cambridge, UK) for the pcDNA3.1-Myc/His-SIRT1 plasmid, Dr. Cho (Yeonsei University, Seoul, Korea) for the p3xFlag-OGT plasmid, Dr. D.J. Vocadlo for the pMal-c2x-OGT plasmid, Dr. Suat Özbek (University of Heidelberg, Heidelberg, Germany) for pGEX-2T-OGT plasmid. This work was supported by the NSFC-Shandong Joint Fund for Marine Science Research Centers (No. U1606403), the Scientific and Technological Innovation Project Financially Supported by Qingdao National Laboratory for Marine Science and Technology (No. 2015ASKJ02), NSFC (No. 81272264), and also a grant from the Fundamental Research Funds for the Central Universities (No. 201562028).

## Author contributions

W.Y. and Y.G. designed, directed, and coordinated the project. C.H., Y.G. and H.S. performed the experiments except where otherwise noted. W.M. and Y.G. discovered SIRT1 was O-GlcNAcylated. J.S. and M.S. determined the enzyme kinetic parameters of SIRT1. J.S. established the stable cell lines with endogenous SIRT1 knockdown and exogenously expressed SIRT1. X.Z. constructed the retroviral SIRT1 expression vectors. X.L. performed the MS analyses; M.S. and F.H. performed protein purification; G.H. performed statistical analysis. W.Y. and Y.G. wrote the manuscript.

## Additional information

**Competing interests:** The authors declare no competing financial interests.

