## [Peer Review File · Nature Communications]

Reviewers' Comments:

Reviewer #1 (Remarks to the Author)

The paper "O-GlcNAcylation of SIRT1" shows that SIRT1 is thus modified on S549 by OGT in response to stressors, such as DNA damage, oxidative damage or fasting. This modification seems to increase SIRT1 enzymatic activity, resulting in a down regulation of p53 and apoptosis in response to the stressors. The data is generally convincing, although I have specific comments below. This modification of SIRT1 has not been previously observed, and the functional effects conferred by are interesting and make this paper an attractive candidate for publication in Nat. Comm.

Specific comments.

1. The GalT assay for O-GlcNAcylation is not described well and why it is specific for this modification is not clear to me. The controls in Fig. 2 increase confidence in the assay, but the authors need to explain and justify it better in Results, since it figures so prominently in later experiments.
2. There are SIRT1 enzymatic assays in Fig. 4, 5, and 7. The authors have apparently purified SIRT1 by IP for these assays. They must show a protein gel and quantification for SIRT1 levels in samples used in each lane of the assay bar graphs. This is crucial.
3. In Fig. 5C, there is a minus under "fasting" in lane 9 that should be a plus. Fig. S3 has the same problem.
4. I do not understand the use of MG132 in the experiments in Fig. 6.
5. I am quite sure the pluses and minuses in Fig. 6E are screwed up. For example, lanes 8 and 11 have identical designations. As it stands, the figure does not make sense. Would like to see the fixed version.
6. In line 57 of Discussion there is a reference 57, which does not exist.
7. The authors should probably reference the finding that SIRT1 is an NAD⁺ dependent deacetylase.

With the above comments properly addressed, this paper would merit publication in Nat. Comm.

Reviewer #2 (Remarks to the Author)

In this study the authors present data demonstrating that OGT associates with SirT1 and that this results in O-GlcNAcylation of SirT1 at Ser 549. The role of O-GlcNAc on SirT1 is assessed using acetylation of p53 as well as a fluorogenic substrate, and suggest that O-GlcNAc activates SirT1 during injury to inhibit apoptosis. While some of the data presented is compelling, there are some critical controls missing and the experimental approach makes it challenging to assess how critical SirT1 O-GlcNAcylation is to stressed cells.

In Figure 2A - OGT can in vitro glycosylate itself. Thus, a common control is to perform the described reaction in the absence of protein/peptide substrate. Given the close molecular weights of OGT and SirT1, adding a OGT + UDP-GlcNAc is critical.

To assess the significance of SirT1 O-GlcNAcylation to cells it is critical to assess what percentage of the SirT1 population is O-GlcNAc modified. There are two potential approaches which utilize click chemistry that can be used to address this concern. Firstly, rather than using Click Chemistry to attach Biotin to GalNAz, the authors could attach polyethylene glycol (Rexach et al, 2010, PMID:20657584). These samples could then be separated by SDS-PAGE, and SirT1 detected. For each site of O-GlcNAcylation SirT1 would move ~4kDa, and the percentage of each glycoform could be quantified. Secondly, instead of labeling SirT1 on beads the authors could label cell lysate with GalT/GalNAz, click on biotin, and then enrich the biotinylated proteins using streptavidin affinity chromatography. Of note, it is critical that the streptavidin affinity chromatography enrich

all biotinylated proteins. Using a dilution series of cell lysate, the authors could assess the percentage of SirT1 that is modified.

One concern raised by the experimental approach taken in this paper is that the authors are detecting O-GlcNAcylation of an associated protein, for instance OGT. While mutation of Ser549 partially addresses this concern, mutation could block association with the co-precipitating protein. To address this concern, the authors should label cell lysate with GalT/GalNAz, click on biotin, and then enrich the biotinylated proteins using streptavidin affinity chromatography and then detect SirT1.

In Figure 3b - it would be helpful if the authors probed the IP to determine if the mutation of Ser549 alters the association between OGT and SirT1.

Figure 4B - This experiment should also include a p53 + TMG condition. This will enable to authors to determine if TMG associated deacetylation is dependent on the overexposed SirT1.

Relevant to all assays: 1) The authors need to state in the methods the amount of NAD used in these assays and the amount of protein (SirT1) used in each assay. 2) It is critical for the authors to demonstrate that the levels of protein are equal in the assays, especially since many of these assays are performed on beads. 3) Relevant to all assays from E.Coli expressed protein - has the SirT1 been separated from OGT? has the unmodified and modified SirT1 been separated?

This study would be significantly strengthened if the authors provided more quantitative data on SirT1 activity; what is the activity in mol/min/mg. In addition, how does O-GlcNAc alter the activity of SirT1? Is the affinity of the enzyme for NAD or the peptide altered by O-GlcNAc? or it is the catalytic efficiency of the enzyme that is altered.

Ultimately, this paper leaves a critical question unanswered. How important is SirT1 O-GlcNAcylation to cellular function. The cleanest way to assess this is to suppress the expression of wild-type SirT1 and re-express the mutant at endogenous levels. This can be done with shRNA or there is an inducible SirT1 mouse. This would enable the authors to cleanly assess the function of glycoSirT1 on cell death, rather than the overexpression assays used in these studies. Moreover, it would enable the authors to look at pan-acetylation in these cells and the acetylation of other SirT1 targets. This is particularly important as for instance acetylation of HSF1 is considered pro survival as it activates this transcription factor.

Minor Issues:

Figure 3A: The full length unmodified peptide is labeled as doubly charged, it is singly charged. 921.88 is the doubly charged species.

In Supplemental Figure 1, 144.08 is labelled as a B3+++ - this ion is more likely a GlcNAc cleavage fragment. It would also be helpful in this figure to indicate that the majority of the masses arise from the Unmodified peptide, for example Y16 - GlcNAc

In Figure 5B - Lanes 8 and 9, are these duplicates or was one of these treated with Fasting.

Figure 6B/C would be more convincing if the authors recapitulated the experiments in the absence of p53 as in 6D.

In Figure 6D - there appear to be some issues with the labeling of this Figure. Lanes 1 and 4 are apparently identical, except that there is cleaved PARP in lane 4. This is also true of 1 and 5, as well as 2 and 6. In fact in 6D, I believe that the Etoposide has been mislabelled.

Supplemental Figure 3, I think another of the Wildtypes is not labeled as fasted

Citation 57 is missing (line 356)

Reviewer #3 (Remarks to the Author)

This manuscript identified a novel modification, the O-GlcNAcylation of SIRT1, which in turn regulates the deacetylase activity of SIRT1. The authors tried to elucidate the function of this modification and found that O-GlcNAcylation of SIRT1 is induced by different stimulus, thus activating SIRT1 to protect cells from apoptosis through deacetylation of p53. The story is interesting, however, it is not appropriate for publication in Nature Communications due to the following reasons. First, although the O-GlcNAcylation of SIRT1 is a novel modification, but the dynamic regulatory machinery has not been fully explored by the authors. Second, the role of SIRT1 as a stress sensor and protecting cells from stress has been extensively studied, yet the authors have not found any novel functions of this novel modification. Finally, the authors identified that the O-GlcNAcylation of SIRT1 regulates its deacetylase activity, which influences the acetylation of p53. However, judged by the data presented, O-GlcNAcylation dead mutant of SIRT1 only loses partial deacetylase activity, which suggests additional regulatory mechanisms. Overall, this manuscript has not reached the standards of Nature Communications.

Point-by-point response to referees' comments

I deeply appreciate the efforts of you, and the comments and suggestions are very constructive. We have completed the experiments for addressing the comments. The point-by-point responses are listed below.

Reviewer #1 (Remarks to the Author):

The paper "O-GlcNAcylation of SIRT1 " shows that SIRT1 is thus modified on S549 by OGT in response to stressors, such as DNA damage, oxidative damage or fasting. This modification seems to increase SIRT1 enzymatic activity, resulting in a down regulation of p53 and apoptosis in response to the stressors. The data is generally convincing, although I have specific comments below. This modification on SIRT1 has not been previously observed, and the functional effects conferred by are interesting and make this paper an attractive candidate for publication in Nat. Comm.

Specific comments.

1. The GalT assay for O-GlcNAcylation is not described well and why it is specific for this modification is not clear to me. The controls in Fig. 2 increase confidence in the assay, but the authors need to explain and justify it better in Results, since it figures so prominently in later experiments.

Answer: According to the reviewer's suggestion, we rewrote this part. Additionally, we also provided an additional result (Figure 2e) in order to demonstrate the specificity of GalT assay for SIRT1 O-GlcNAcylation.

2. There are SIRT1 enzymatic assays in Fig. 4, 5, and 7. The authors have apparently purified SIRT1 by IP for these assays. They must show a protein gel and quantification for SIRT1 levels in samples used in each lane of the assay bar graphs. This is crucial.

Answer: As the reviewer pointed, we quantified SIRT1 levels while examining SIRT1 activity. The SIRT1 samples from cultured cells and tissues were determined by WB, while recombinant SIRT1 proteins from *E.coli* were analysed on SDS-PAGE followed by coomassie brilliant blue staining. We have added the figures for quantification of SIRT1 levels.

3. In Fig. 5C, there is a minus under "fasting" in lane 9 that should be a plus. Fig. S3 has the same problem.

Answer: We feel sorry for our mistakes, and we have revised the figures.

- I do not understand the use of MG132 in the experiments in Fig. 6.

Answer: The purpose of the experiment is to analyse the SIRT1-mediated deacetylation of p53 under normal conditions and etoposide treatment conditions. However, under normal culture conditions, the half life of p53 protein was very short because of the degradation by proteasome; however, p53 protein was stabilized under etoposide treatment conditions. To normalize the levels of p53 proteins under both conditions, proteasome inhibitor MG132 was added to cell cultures prior to introducing stress stimuli. MG132 was also used in other papers for similar purposes.

- I am quite sure the pluses and minuses in Fig. 6E are screwed up. For example, lanes 8 and 11 have identical designations. As it stands, the figure does not make sense. Would like to see the fixed version.

Answer: We feel sorry for our mistake. Considering the suggestions of another reviewer, we have repeated the cell apoptosis experiment using the cells expressing wild-type or mutant SIRT1 while suppressing endogenous SIRT1 via shRNA. In the latest results (Figure S11A), we have corrected this error.

- In line 57 of Discussion there is a reference 57, which does not exist.

Answer: Thanks very much, we have deleted the reference.

- The authors should probably reference the finding that SIRT1 is an NAD⁺ dependent deacetylase.

Answer: Thanks for the suggestion, we have added the references (*Nature*. 2000 Feb17;403(6771):795-800; *Proc Natl AcadSci U S A*. 2000 Jun 6;97(12):6658-63).

With the above comments properly addressed, this paper would merit publication in Nat. Comm.

Reviewer #2:

In this study the authors present data demonstrating that OGT associates with SirT1 and that this results in O-GlcNAcylation of SirT1 at Ser 549. The role of O-GlcNAc on SirT1 is assessed using acetylation of p53 as well as a fluorogenic substrate, and suggest that O-GlcNAc activates SirT1 during injury to inhibit apoptosis. While some

of the data presented is compelling, there are some critical controls missing and the experimental approach makes it challenging to assess how critical SirT1 O-GlcNAcylation is to stressed cells.

1. In Figure 2A - OGT can *in vitro* glycosylate itself. Thus, a common control is to perform the described reaction in the absence of protein/peptide substrate. Given the close molecular weights of OGT and SirT1, adding a OGT + UDP-GlcNAc is critical.

Answer: According to the reviewer's suggestion, a negative control containing OGT and UDP-GlcNAc but not SIRT1 was performed. The results indicated that IB with anti-O-GlcNAc antibody (RL2) can detect the O-GlcNAcylation of SIRT1 but not OGT.

Although it is reported that OGT can O-GlcNAcylate itself *in vitro* and is detectable by IB with RL2, we can only detect O-GlcNAcylation of SIRT1 but not OGT, except for overexposure. This result suggests that SIRT1 is more readily O-GlcNAcylated than OGT itself *in vitro*.

Additionally, OGT used in our assay was fused to MBP tag, so the molecular weight of OGT (apparent molecular weight 160 kDa) was much higher than His-fused SIRT1 (apparent molecular weight 130 kDa). Altogether, these results demonstrate that SIRT1 is indeed O-GlcNAcylated *in vitro*.

2. To assess the significance of SirT1 O-GlcNAcylation to cells it is critical to assess what percentage of the SirT1 population is O-GlcNAc modified. There are two potential approaches which utilize click chemistry that can be used to address this concern. Firstly, rather than using Click Chemistry to attach Biotin to GalNAz, the authors could attach polyethylene glycol (Rexach et al, 2010, PMID:20657584). These samples could then be separated by SDS-PAGE, and SirT1 detected. For each site of O-GlcNAcylation SirT1 would move ~4kDa, and the percentage of each glycoform could be quantified. Secondly, instead of labeling SirT1 on beads the authors could label cell lysate with GalT/GalNAz, click on biotin, and then enrich the biotinylated proteins using streptavidin affinity chromatography. Of note, it is critical that the streptavidin affinity chromatography enrich all biotinylated proteins. Using a dilution series of cell lysate, the authors could assess the percentage of SirT1 that is modified.

Answer: Thanks for reviewer's suggestion. Because the reagents for the first

suggested method are not commercially available for us, the second method was used to assess what percentage of the SIRT1 population is O-GlcNAcylated.

As the reviewer suggested, we labelled cell lysate with GalNAz, click on biotin, and then enrich the biotinylated proteins using streptavidin affinity chromatography (Figure 6D). At the same time, a dilution series of cell lysate were immunoblotted with anti-SIRT1 antibody, and a standard curve was generated (Supplementary Fig. 13). Based on this standard curve, the percentage of O-GlcNAcylated SIRT1 was about 4% under basal condition, but increased to about 12% after etoposide treatment, suggesting that O-GlcNAcylation of SIRT1 might only involve in some of SIRT1's functional complexes.

3. One concern raised by the experimental approach taken in this paper is that the authors are detecting O-GlcNAcylation of an associated protein, for instance OGT. While mutation of Ser549 partially addresses this concern, mutation could block association with the co-precipitating protein. To address this concern, the authors should label cell lysate with GalT/GalNAz, click on biotin, and then enrich the biotinylated proteins using streptavidin affinity chromatography and then detect SirT1.

Answer: Thanks for the suggestion, we have detected two SIRT1-associated proteins, SIRT1 negative regulator DBC1 and OGT, and the results indicated that neither DBC1 nor OGT was detectable in the precipitates (Supplementary Fig. 1). Additionally, we also carried out the experimental program for chemoenzymatic labelling as suggested (Fig. 2e).

4. In Figure 3b - it would be helpful if the authors probed the IP to determine if the mutation of Ser549 alters the association between OGT and SirT1.

Answer: We have detected the interaction of wSIRT1/S549A with OGT by CoIP and Pull-down assay, the results showed that the mutation of Ser549 can't alter the association between OGT and SIRT1 (Fig 3d and Supplementary Fig. 3).

5. Figure 4B - This experiment should also include a p53 + TMG condition. This will enable to authors to determine if TMG associated deacetylation is dependent on the overexposed SirT1.

Answer: Thanks for the suggestion, we have done the experiments. The results showed that TMG could not modulate the acetylation status of p53 in the absence of SIRT1 under our experiment condition (Figure 6A), supporting our original

conclusion.

6. Relevant to all assays: 1) The authors need to state in the methods the amount of NAD used in these assays and the amount of protein (SirT1) used in each assay. 2) It is critical for the authors to demonstrate that the levels of protein are equal in the assays, especially since many of these assays are performed on beads. 3) Relevant to all assays from *E. coli* expressed protein - has the SirT1 been separated from OGT? has the unmodified and modified SirT1 been separated?

Answer: Thanks for the suggestion.

1) We have added the detail information for SIRT1 activity assay in the methods section. SIRT1 activity assay was carried out with 25 μ M fluorogenic acetylated p53 peptide substrate and 500 μ M NAD⁺ (or the indicated concentrations of fluorogenic acetylated peptide substrate and NAD⁺ as described in the manuscript) for 20 min at 37 °C.

2) When the SIRT1 activity assays were performed, the levels of SIRT1 protein were also quantified and the SIRT1 activity was normalized by SIRT1 protein level. We have added the results of WB or coomassie blue staining for SIRT1 proteins.

3) As referee's comments, because of the interaction of OGT and SIRT1, visible OGT proteins could be co-purified with SIRT1 from OGT and SIRT1 co-expressed *E. coli*. In order to eliminate the effects of OGT protein, the recombinant SIRT1 was purified by HisTrap HP column, and then the eluted protein samples were loaded on to the MBPTrap HP column for removing the co-purified MBP-OGT. OGT was effectively eliminated from purified SIRT1 protein (Figure 4).

7. This study would be significantly strengthened if the authors provided more quantitative data on SirT1 activity; what is the activity in mol/min/mg. In addition, how does O-GlcNAc alter the activity of SirT1? Is the affinity of the enzyme for NAD or the peptide altered by O-GlcNAc? or it is the catalytic efficiency of the enzyme that is altered.

Answer: According to the suggestion, wild-type SIRT1 and mutant SIRT1^{S549A} were co-expressed with/without OGT in *E. coli*. Then, wtSIRT1-GlcNAc, SIRT1^{S549A}-GlcNAc, wtSIRT1 and mutant SIRT1^{S549A} were purified, and their purity and O-GlcNAcylation status were determined.

The specific activity and the kinetic parameters of SIRT1 for acetylated p53 peptide substrate were analysed. The data showed that O-GlcNAcylation enhanced substrate affinity (K_m) and catalytic efficiency (k_{cat}/K_m) of wtSIRT1 much more than SIRT1^{S549A} (Fig4).

Due to the labile, dynamic, and substoichiometric characteristics of O-GlcNAc, it is technically impossible to obtain homogeneously O-GlcNAcylated protein to our knowledge. In this study, we used a variety of means to obtain highly O-GlcNAcylated recombinant SIRT1, including the addition of OGA inhibitors in all used buffers, keeping protein samples at low temperature condition and completing the experiment within 16 h. Despite these efforts, we found that the O-GlcNAcylation level of SIRT1 was still obviously reduced after purification. We also attempted to assess what the percentage of the SIRT1 population was O-GlcNAcylated (the assay was carried out as Figure 6d), but we found that the O-GlcNAcylation level of SIRT1 was markedly reduced after completing the assessment process (about 3 days). Therefore, the data (about 30%) were not credible and were not provided in the manuscript.

Altogether, these data demonstrate that O-GlcNAcylation of SIRT1 indeed plays important roles for SIRT1 activity and the effect might be underestimated due to the labile property of O-GlcNAcylation.

8. Ultimately, this paper leaves a critical question unanswered. How important is SirT1 O-GlcNAcylation to cellular function. The cleanest way to assess this is to suppress the expression of wild-type SirT1 and re-express the mutant at endogenous levels. This can be done with shRNA or there is an inducible SirT1 mouse. This would enable the authors to cleanly assess the function of glycoSirT1 on cell death, rather than the overexpression assays used in these studies. Moreover, it would enable the authors to look at pan-acetylation in these cells and the acetylation of other SirT1 targets. This is particularly important as for instance acetylation of HSF1 is considered pro survival as it activates this transcription factor.

Answer: Thanks for the constructive suggestions. According to the suggestions, we silenced the endogenous SIRT1 and reintroduced exogenous SIRT1 (wild-type or mutant SIRT1) by lentiviral infection in NCI-H1299 cells. The expression levels of exogenous SIRT1 were comparable to that of endogenous SIRT1 (Figure S9). The cell lines were used to assess the function of SIRT1 O-GlcNAcylation on

cell death/apoptosis (Figure 6).

As referee's suggestion, we also detected the pan-acetylation in these cells, but there were no obviously difference, even in the SIRT1 silenced cells (data not shown), which may be due to the presence of multiple protein deacetylases in mammalian cells.

Considering that SIRT1 can deacetylate a wide range of substrate proteins, some of them, such as HSF1 and FOXO3, can also mediate the cytoprotective role of SIRT1. We also detected the acetylation of HSF1 as referee's suggestion in NCI-H1299 cells, but it was undetectable in our experiment. We did not try to explain the reason for this. However, we found that O-GlcNAcylation of SIRT1 enhanced the deacetylation for FOXO3 after etoposide treatment, which was consistent with the results of p53.

Minor Issues:

1. Figure 3A: The full length unmodified peptide is labeled as doubly charged, it is singly charged. 921.88 is the doubly charged species.

Answer: Thanks, we have corrected the mistake.

2. In Supplemental Figure 1, 144.08 is labelled as a B3+++ - this ion is more likely a GlcNAc cleavage fragment. It would also be helpful in this figure to indicate that the majority of the masses arise from the Unmodified peptide, for example Y16 - GlcNAc

Answer: We feel sorry for our mistake, it has been corrected.

3. In Figure 5B - Lanes 8 and 9, are these duplicates or was one of these treated with Fasting.

Answer: We feel sorry for our mistake. Based on our new findings that stress stimuli rapidly induced the O-GlcNAcylation of SIRT1 in NCI-H1299 cells in a highly dynamic time-dependent manner, we repeated the experiments of Figure 5B at a new condition and the current results were more convincing.

4. Figure 6B/C would be more convincing if the authors recapitulated the experiments in the absence of p53 as in 6D.

Answer: This suggestion is good. However, we just wanted to determine the functional difference between wild-type and mutant SIRT1 by detecting p53

transcriptional activity. For this purpose, we think that the original experiments can explain the problem.

5. In Figure 6D - there appear to be some issues with the labeling of this Figure. Lanes 1 and 4 are apparently identical, except that there is cleaved PARP in lane 4. This is also true of 1 and 5, as well as 2 and 6. In fact in 6D, I believe that the Etoposide has been mislabelled.

Answer: We feel sorry for our mistake. Based on the above comments, we repeated the experiment with the newly constructed cell line (Figure S11a).

6. Supplemental Figure 3, I think another of the Wildtypes is not labeled as fasted

Answer: We feel sorry for our mistake, and we have revised the Figure S6 (original Figure S3).

7. Citation 57 is missing (line 356)

Answer: Thanks, we have revised the sentence.

Reviewer #3 (Remarks to the Author):

This manuscript identified a novel modification, the O-GlcNAcylation of SIRT1, which in turns regulates the deacetylase activity of SIRT1. The authors tried to elucidate the function of this modification and found that O-GlcNAcylation of SIRT1 is induced by different stimulus, thus activating SIRT1 to protect cells from apoptosis through deacetylation of p53. The story is interesting, however, it is not appropriate for publication in Nature Communications due to the following reasons. First, although the O-GlcNAcylation of SIRT1 is a novel modification, but the dynamic regulatory machinery has not been fully explored by the authors. Second, the role of SIRT1 as a stress sensor and protecting cells from stress has been extensively studied, yet the authors have not found any novel functions of this novel modification. Finally, the authors identified that the O-GlcNAcylation of SIRT1 regulates its deacetylase activity, which influences the acetylation of p53. However, judged by the data presented, O-GlcNAcylation dead mutant of SIRT1 only losses partial deacetylase activity, which suggests additional regulatory mechanisms. Overall, this manuscript has not reached the standards of Nature Communications.

Answer: Thanks for the comments. In the revised manuscripts, we showed that stress stimuli rapidly induced the O-GlcNAcylation of SIRT1 in NCI-H1299 cells in a highly dynamic time-dependent manner (Figure 5A). These results strongly supported

the roles of SIRT1 O-GlcNAcylation in cellular stress response. We also assessed what percentage of the SIRT1 population is O-GlcNAc modified. The result showed that the percentage of O-GlcNAcylated SIRT1 was about 4% under basal condition, but increased to about 12% after etoposide treatment (Fig 6d), suggesting that O-GlcNAcylation of SIRT1 only participated in a small number of SIRT1's functional complexes. However, it is another meaningful and burdensome work to fully elucidate the dynamic regulatory machinery for SIRT1 O-GlcNAcylation, which is not the essential work for this study.

To fully investigate the role of O-GlcNAcylation in SIRT1 activity, we purified the recombinant SIRT1 with/without O-GlcNAcylation and examined the enzyme kinetic parameters *in vitro*. The enzyme kinetic assays showed that O-GlcNAcylation obviously enhanced substrate affinity (Km) and catalytic efficiency (kcat/Km) of wtSIRT1 for acetylated p53 peptide (Fig 4e and f). However, the effects of O-GlcNAcylation on SIRT1^{S549A} were not as obvious as that on wtSIRT1 (Fig 4e and f). These results demonstrate that O-GlcNAcylation at S549 is important for the regulation of catalytic efficiency and substrate affinity of SIRT1. To study the effect of O-GlcNAcylation on SIRT1 deacetylase activity *in vivo*, the acetylation status of p53 was monitored in the original version of the manuscript (Fig 4a), and the acetylation of histone H3 was also detected in this revised manuscript (Fig 4b). All the results indicate that O-GlcNAcylation of SIRT1 enhance its deacetylase activity both *in vitro* and *in vivo*.

To fully assess how important is SIRT1 O-GlcNAcylation to cellular function, we silenced the endogenous SIRT1 and reintroduced exogenous SIRT1 (wild-type or mutant SIRT1) by lentiviral infection in NCI-H1299 cells in the revise version. The expression levels of exogenous SIRT1 were comparable to that of endogenous SIRT1 (Figure S9). The cell lines were used to assess the function of SIRT1 O-GlcNAcylation on cell death/apoptosis (Figure 6). In the current version of manuscript, the effects of SIRT1 O-GlcNAcylation are more pronounced when eliminating endogenous SIRT1 by shRNA.

In this study, we found that the O-GlcNAcylation of SIRT1 obviously enhanced the activity of SIRT1, but not necessary for SIRT1 deacetylase activity. Therefore, "O-GlcNAcylation dead mutant of SIRT1 only losses partial deacetylase activity" is consistent with our main findings.

In addition to the above issues, we also added some other important results in the

revised manuscript. We believe that the revised manuscript will be more interesting.

Reviewers' Comments:

Reviewer #1:

Remarks to the Author:

The revised paper has addressed my concerns. The biological effects of O-GlcNAc modification of SIRT1 are modest, but this paper is more about establishing that the modification exists and is stimulated by DNA damage. I suspect there is a not yet determined condition in which the modification is most important.

Reviewer #2:

Remarks to the Author:

This study demonstrates that SirT1 is O-GlcNAc modified during stress, and that glycosylation of SirT1 results in activation of the deacetylase activity thus promoting cell survival. In the past review, the authors were asked to address specificity of the detection technique, provide additional experimental details, assess the impact of O-GlcNAcylation on enzymatic parameters of SirT1 to determine the mechanisms by which O-GlcNAc is resulting in altered SirT1 activity, and to determine the physiological significance of SirT1 O-GlcNAcylation.

For the most part the authors have addressed my concerns. However, mutation of the predominant SirT1 O-GlcNAcylation site has only a modest effect on basal and stress-induced acetylation (Figure 4, 6), and a modest impact on Survival (Figure 6). Quantifying the impact on acetylation is critical. As suggested in my previous review, I feel that examining pan-acetylation would be helpful.

The data in Figure 4 contradict each other. Figure 4C, performed on IPs, shows a two fold increase in activity upon TMG treatment. However, in figure 4F there is an almost two fold increase in activity of both the WT and Glyco-mutant in response to co-expression with OGT. Moreover, in spite of the significant difference in O-GlcNAcylation in Figure 4D, there is little difference between WT-GlcNAc Sirt1 (7.2 nmol/min/mg) and Ser549A-GlcNAc as 7.5 nmol/min/mg. Thus, the authors need to address this discrepancy.

Reviewer #3:

Remarks to the Author:

The authors have addressed the major concerns in an acceptable way. The authors have further explored the detailed dynamics of the O-GlcNAcylation of SIRT1 upon stress and its function in regulating p53 activity.

Responses to Reviewers' Comments

We greatly appreciate the constructive reviews provided by the reviewers. Below, please find our point-by-point responses to questions and comments. We have revised the manuscript according to the reviewers' questions and format requirements.

Reviewer #1:

The revised paper has addressed my concerns. The biological effects of O-GlcNAc modification of SIRT1 are modest, but this paper is more about establishing that the modification exists and is stimulated by DNA damage. I suspect there is a not yet determined condition in which the modification is most important.

Thanks for your positive comments.

Reviewer #2:

This study demonstrates that SirT1 is O-GlcNAc modified during stress, and that glycosylation of SirT1 results in activation of the deacetylase activity thus promoting cell survival. In the past review, the authors were asked to address specificity of the detection technique, provide additional experimental details, assess the impact of O-GlcNAcylation on enzymatic parameters of SirT1 to determine the mechanisms by which O-GlcNAc is resulting in altered SirT1 activity, and to determine the physiological significance of SirT1 O-GlcNAcylation.

For the most part the authors have addressed my concerns. However, mutation of the predominant SirT1 O-GlcNAcylation site has only a modest effect on basal and stress-induced acetylation (Figure 4, 6), and a modest impact on Survival (Figure 6). Quantifying the impact on acetylation is critical. As suggested in my previous review, I feel that examining pan-acetylation would be helpful.

We would like to thank this reviewer for asking this question. We have examined the pan-acetylation, but there was no obvious difference between the wild-type and S549A mutant transfected cells. We hypothesize that this may be due to the presence of large amounts of acetylated proteins and multiple deacetylases in the cells, whereas SIRT1 can only deacetylate a small amount of the acetylated proteins and that some of the target proteins of SIRT1 may also be substrates for other deacetylases. Consistent with our speculation, we found that over-expression or silencing SIRT1 can't obviously change the pan-acetylation status. In order to exclude the possibility

of antibody preference, we used two different pan-acetylation antibodies (Cell Signaling Technology, #CST-9441; PTM Biolabs, #PTM-101) in this study.

In order to evaluate the deacetylation activity of SIRT1 *in vivo*, we further examined the acetylation level of two other target proteins (FOXO3 and H3) of SIRT1 (Fig. 4b and Fig. 6c) in addition to p53. The results strongly support our findings that O-GlcNAcylation enhances the deacetylase activity of SIRT1 *in vivo*.

As the reviewer pointed, “*mutation of the predominant SirT1 O-GlcNAcylation site has only a modest effect on basal and stress-induced acetylation (Figure 4, 6), and a modest impact on Survival (Figure 6)*”. However, even the expression of wtSIRT1 only modestly reduced the acetylation levels of p53 and H3 (Fig. 4a and b, lane 3). Therefore, we think this does not mean that the effect of SIRT1 O-GlcNAcylation is weak, because the effect of the SIRT1 protein itself is modest in this respect. In this study, we chose a relatively low concentration of etoposide to induce cell apoptosis/death, so the percentage of cell death modestly increased from about 8% (background) to about 17% (Fig. 6b). We also found that SIRT1 knockdown increased cell death (from 17% to 24%), which was consistent with the previous study (Cell, 2001, Vol. 107, 137–148, Figure 5B). Re-expression of wild-type SIRT1 completely rescues the cell survival ability, but the effect of S549A expression is less than half of cellular protective effect of wild-type SIRT1 (Fig. 6b and Supplementary Fig. 12b). Collectively, these data indicated that O-GlcNAcylation at S549 site of SIRT1 plays important roles for the activation of SIRT1 and its cellular protective effects during stress response.

The data in Figure 4 contradict each other. Figure 4C, performed on IPs, shows a two fold increase in activity upon TMG treatment. However, in figure 4F there is an almost two fold increase in activity of both the WT and Glyco-mutant in response to co-expression with OGT. Moreover, in spite of the significant difference in O-GlcNAcylation in Figure 4D, there is little difference between WT-GlcNAc Sirt1 (7.2 nmol/min/mg) and Ser549A-GlcNAc as 7.5 nmol/min/mg. Thus, the authors need to address this discrepancy.

I am sorry that we did not provide the detailed information in the original manuscript, which was added to the current version. In figure 4C, the SIRT1 proteins were immunopurified from cell extracts using anti-FLAG M2 Affinity Gel, and the relative deacetylase activity was determined *in vitro* at the concentration of 25 μ M fluorogenic acetylated p53 peptide substrate and 500 μ M NAD⁺. In Figure 4F, SIRT1 was coexpressed with OGT in *E. coli* and purified, and then the specific deacetylase activity (nmol/min/mg) of SIRT1 was determined at the concentration of 250 μ M fluorogenic acetylated p53

peptide substrate and 500 μM NAD^+ .

The seemingly contradictory results are in fact consistent with our findings. The enzyme kinetic assays showed that O-GlcNAcylation obviously enhanced substrate affinity (K_m) of wtSIRT1 for acetylated p53 peptide. Therefore, O-GlcNAc enhanced the activity of SIRT1 at low concentration of acetylated p53 peptide substrate (25 μM), but O-GlcNAc did not have an obvious effect at a nearly saturated substrate concentration (250 μM fluorogenic acetylated p53 peptide substrate). Additionally, in the original version of our manuscript, we showed the similar effect of O-GlcNAc on the relative deacetylase activity of SIRT1 proteins from both H1299 cells and *E. coli* at the concentration of 25 μM fluorogenic acetylated p53 peptide substrate and 500 μM NAD^+ .

Reviewer #3:

The authors have addressed the major concerns in an acceptable way. The authors have further explored the detailed dynamics of the O-GlcNAcylation of SIRT1 upon stress and its function in regulating p53 activity.

Answer: Thanks for your positive comments.

Reviewers' Comments:

Reviewer #2:

Remarks to the Author:

I thank the authors for their careful consideration of my comments. While I find most of their responses satisfactory, I am still challenged by the data in what was Figure 4, now supplemental Figure 5 and supplemental Table 1. For me, these data are critical as cellular data is complicated by observations that SirT1 can be phosphorylated at Ser549.

I agree, that the author's data supports a change in the affinity of SirT1 for its substrate when O-GlcNAc modified. However, the authors should not observe a two-fold increase in mSirT1 activity when co-expressed with OGT (5.1 nmol/min/mg to 7.2nmol/min/mg). These data suggest that there are additional sites of O-GlcNAcylation on SirT1 (as supported by Supplemental figure 5) and that these sites are more critical for SirT1 activity than Ser 549.

We greatly appreciate the reviewer's questions and suggestions. It is helpful to deepen our understanding of some key issues. And thanks the editors very much for giving us the opportunity to answer the questions. Below, please find our point-by-point responses to questions and comments, which is also provided as an attachment.

REVIEWERS' COMMENTS:

Reviewer #2 (Remarks to the Author):

Report 1:

I thank the authors for their careful consideration of my comments. While I find most of their responses satisfactory, I am still challenged by the data in what was Figure 4, now supplemental Figure 5 and supplemental Table 1. For me, these data are critical as cellular data is complicated by observations that SirT1 can be phosphorylated at Ser549.

Answer: Thanks for your comments, I agree with your opinion. The cellular data is complicated because of other post-translational modifications. Therefore, the data from the *in vitro* O-GlcNAcylated SIRT1 is critical for illustrating the importance of Ser 549 O-GlcNAcylation. In the answer to the following question, I will explain and discuss our results in detail.

I agree, that the author's data supports a change in the affinity of SirT1 for its substrate when O-GlcNAc modified. However, the authors should not observe a two-fold increase in mSirT1 activity when co-expressed with OGT (5.1 nmol/min/mg to 7.2nmol/min/mg). These data suggest that there are additional sites of O-GlcNAcylation on SirT1 (as supported by Supplemental figure 5) and that these sites are more critical for SirT1 activity than Ser 549.

Answer: Thanks for your questions. As you pointed, in addition to the Ser 549 site, we also identified other O-GlcNAcylated sites or peptides in SIRT1 when co-expressed with OGT in *E. coli* (in the table and figures below).

O-GlcNAcylated peptide/site	Detection methods	Mutation
VRPVALIPSSIPHEVPQILINR	ETD MS (Ser 454 is modified)	Ser 454 was mutated to Ala (S454A).
ELAYLSELPPPTPLHVSEDSSSPER	Q-Tof (the peptide is modified)	All the Ser and Thr were mutated to Ala (S/T525-540A)

MS spectrum for O-GlcNAc site mapping

1. VRPVALIPSSIPHEVPQILINR

2. ELAYLSELPTPLHVSEDSSPER

However, only the mutation of Ser 549 obviously decreased the O-GlcNAcylation of SIRT1 in NCI-H1299 cells (in the Figure below), suggesting that other O-GlcNAcylation sites were not critical in cells. One possible explanation is that the binding of other protein partners to SIRT1 hinders the modification of these sites by OGT in cells. This may also be due to the high level of OGT expression in *E. coli*. Therefore, the sites, that are not easily modified in H1299 cells, are O-GlcNAcylated in *E. coli*. The reasons are not important in this study. Importantly, we deduced that Ser 549 was the major and critical O-GlcNAcylation site of SIRT1 in cells. Because the other sites were only O-GlcNAcylated *in vitro*, the results of these sites are not present in the manuscript.

O-GlcNAcylation levels of SIRT1 mutants were detected. The Flag tagged SIRT1 mutants were expressed in NCI-H1299 cells and treated with OGA inhibitor Thiamet-G for 2 h. Their O-GlcNAcylation levels were detected by enzymatic labelling after immunoprecipitated with anti-Flag antibody. Streptavidin (STV) was used for O-GlcNAcylation (O-GlcNAc was labelled by biotin) blotting. Anti-Flag antibody was used to blotting immunoprecipitated SIRT1.

As described above, there are some other O-GlcNAcylation sites on SIRT1 when co-expressed with OGT in *E. coli*. Therefore, the data of the enzyme kinetic assay (Figure 4d and supplemental Table 1) were complicated and seem to be confusion. But the Km data clearly indicated the issues below.

1) O-GlcNAcylation of the wtSIRT1 enhances the affinity for its peptide substrate (88.7 μM and 61.8 μM), but the effect of O-GlcNAcylation on mutant SIRT1 is very slight (60.3 μM and 50.5 μM), indicating the O-GlcNAcylation on Ser 549 increases the affinity for its substrate.

2) The mutation itself (Ser to Ala) visibly enhances the affinity for its peptide substrate (88.7 μM and 61.8 μM), suggesting Ser 549 of SIRT1 is important for substrate binding regulation.

The data of the SIRT1 specific activity seems to be confusion, but we think that this can be interpreted reasonably. The above results showed that the Ser 549 O-GlcNAcylation enhanced the affinity of SIRT1 for its peptide substrate. However, in this assay, we used an almost saturated substrate concentration (250 μM) according the definition of enzyme activity. Under this condition, the role of Ser 549 O-GlcNAcylation in substrate binding was eliminated. O-GlcNAcylation on other sites might enhance SIRT1 activity through other mechanisms. Because O-GlcNAcylation on these sites is undetectable in cells, there is no need for further study on them. We think that this study does not need to provide the specific activity of SIRT1, and the data have been removed from the manuscript. Instead, we have detected the relative activity of purified SIRT1 co-expressed with OGT in *E.coli* at

low concentration of its substrate (25 μ M). As we expected, we observed a nearly 3-fold increase in wtSIRT1 activity when co-expressed with OGT, but only a slight increase in SIRT1^{S549A} when co-expressed with OGT (Supplementary Fig. 5b).

Report 2:

As stated in my previous review; I find no issue with the conclusion that O-GlcNAc moderately enhances the affinity of SirT1 for its substrate. However, for me, the in vitro data is the most critical in the manuscript and indicates presence of an additional glycosylation site that is relevant to this story. The authors argue that while their mass spectrometry data indicates an additional O-GlcNAc modified peptide, only mutation of Ser549 impacts O-GlcNAcylation in vivo.

There are several possible explanations for this: 1) their mass spectrometry did not detect an additional relevant O-GlcNAc modified peptides. I note that the authors declined an experiment in the initial review using PEG addition that would have addressed the number of total sites on SirT1; and 2) There are clearly sites modified by OGT in vitro that are not modified in vivo at high stoichiometry, as evidenced by the ability of the authors to detect in vitro O-GlcNAc modified SirT1 with RL2 but not in vivo

O-GlcNAc modified SirT1. Again, I note that the method used to detect SirT1 (Gal-T labeling, followed by Click chemistry) is not without bias, and that it is possible that another site exists in the Ser549Ala mutant. The authors then present additional activity data, using a sub-saturating concentration of NAD, in which the difference in activity between glyco-WT SirT1 and glyco-Ser549A Sirt1 is compared. I note, that while the difference in activity between WT and Mutant Sirt1 is now more significant (3 fold versus 1.8 fold), that there is still an increase in activity between unmodified Ser549A SirT1 (1.4 fold) or glyco Ser549A Sirt1 (1.8 fold). Thus, I suggest that the authors either present this data and expand their discussion of the potential roles of O-GlcNAcon SirT1 or that the authors deglycosylate SirT1 and demonstrate that activity of the in vitro O-GlcNAc modified SirT1 returns to baseline. Either way, I feel that the authors need to adapt their discussion of their data to reflect more of the excellent points made in their rebuttal (Page 5, line 165).

Answer: Thanks for your comments and suggestion. We have presented the mentioned data as Supplementary Fig. 5b. And we have discussed the concerns raised by reviewer 2 in the manuscript (the third paragraph of Discussion section).